# Improving Variational Autoencoder Estimation from Incomplete Data with Mixture Variational Families

**Vaidotas Simkus** *vaidotas.simkus@ed.ac.uk*
**Michael U. Gutmann** *michael.gutmann@ed.ac.uk*
*School of Informatics*
*University of Edinburgh*

**Reviewed on OpenReview:** *https: // openreview. net/ forum? id= lLVmIvZfry*

## Abstract

We consider the task of estimating variational autoencoders (VAEs) when the training data is incomplete. We show that missing data increases the complexity of the model's posterior distribution over the latent variables compared to the fully-observed case. The increased complexity may adversely affect the fit of the model due to a mismatch between the variational and model posterior distributions. We introduce two strategies based on (i) finite variational-mixture and (ii) imputation-based variational-mixture distributions to address the increased posterior complexity. Through a comprehensive evaluation of the proposed approaches, we show that variational mixtures are effective at improving the accuracy of VAE estimation from incomplete data.

## 1 Introduction

Deep latent variable models, as introduced by Kingma & Welling (2013); Rezende et al. (2014); Goodfellow et al. (2014); Sohl-Dickstein et al. (2015); Krishnan et al. (2016); Dinh et al. (2017), have emerged as a predominant approach to model real-world data. The models excel in capturing the intricate nature of data by representing it within a well-structured latent space. *However, they typically require large amounts of fully-observed data at training time, while practitioners in many domains often only have access to incomplete data sets.*

In this paper we focus on the class of variational autoencoders (VAEs, Kingma & Welling, 2013; Rezende et al., 2014) and investigate the implications of incomplete training data on model estimation. Our contributions are as follows:

- We show that data missingness can add significant complexity to the model posterior of the latent variables, hence requiring more flexible variational families compared to scenarios with fully-observed data (section 3).

- We propose finite variational-mixture approaches to deal with the increased complexity due to missingness for both standard and importance-weighted ELBOs (section 4.1).

- We further propose an imputation-based variational-mixture approach, which decouples model estimation from data missingness problems, and as a result, improves model estimation when using the standard ELBO (section 4.2).

- We evaluate the proposed methods for VAE estimation on synthetic and realistic data sets with missing data (section 6).

The proposed methods achieve better or similar estimation performance compared to existing methods that do not use variational mixtures. Moreover, the mixtures are formed by the variational families that are used

in the fully-observed case, which allows us to seamlessly re-use the inductive biases from the well-studied scenarios with fully-observed data (see e.g. Miao et al., 2022, for the importance of inductive biases in VAEs).

## 2 Background: Standard approach for VAEs estimation from incomplete data

We consider the situation where some part of the training data-points might be missing. We denote the observed and missing parts of the $i$-th data-point $\boldsymbol{x}^i$ by $\boldsymbol{x}^i_{\mathrm{obs}}$ and $\boldsymbol{x}^i_{\mathrm{mis}}$, respectively, where $\boldsymbol{x}^i$ is $D$-dimensional and the dimensions of $\boldsymbol{x}^i_{\mathrm{obs}}$ and $\boldsymbol{x}^i_{\mathrm{mis}}$ must add to $D$. This split into observed and missing components corresponds to a missingness pattern $\boldsymbol{m}^i \in \{0,1\}^D$ with $m^i_j = 1$ if the $j$-th dimension is observed and $m^i_j = 0$ if the dimension is missing. The missingness pattern $\boldsymbol{m}^i$ is generally different for each data-point and is a realisation of a random variable $\boldsymbol{m}$ that follows a typically unknown missingness distribution $p^*(\boldsymbol{m} \mid \boldsymbol{x}^i)$. We make the common assumption that the missingness distribution does not depend on the missing variables, which is known as the ignorable missingness or missing-at-random assumption (MAR, e.g. Little & Rubin, 2002, Section 1.3).[1] The MAR assumption allows us to ignore the missingness pattern $\boldsymbol{m}^i$ when fitting a model $p_{\boldsymbol{\theta}}(\boldsymbol{x})$ of the true distribution $p^*(\boldsymbol{x})$ from incomplete data (see e.g. Seaman et al., 2013, Theorem 1), as well as when performing multiple imputation of the missing data (see e.g. van Buuren, 2018, Section 2.2.6).

The VAE model with parameters $\boldsymbol{\theta}$ is typically specified using a decoder distribution $p_{\boldsymbol{\theta}}(\boldsymbol{x} \mid \boldsymbol{z})$, parametrised using a neural network, and a prior $p_{\boldsymbol{\theta}}(\boldsymbol{z})$ over the latents $\boldsymbol{z}$ that can either be fixed or learnt. A principled approach to handling incomplete training data is then to marginalise the missing variables from the likelihood $p_{\boldsymbol{\theta}}(\boldsymbol{x})$, which yields the marginal likelihood

$$p_{\boldsymbol{\theta}}(\boldsymbol{x}^i_{\mathrm{obs}}) = \int p_{\boldsymbol{\theta}}(\boldsymbol{x}^i_{\mathrm{obs}}, \boldsymbol{x}^i_{\mathrm{mis}})\, \mathrm{d}\boldsymbol{x}^i_{\mathrm{mis}} = \iint p_{\boldsymbol{\theta}}(\boldsymbol{x}^i_{\mathrm{obs}}, \boldsymbol{x}^i_{\mathrm{mis}} \mid \boldsymbol{z}) p_{\boldsymbol{\theta}}(\boldsymbol{z})\, \mathrm{d}\boldsymbol{z}\, \mathrm{d}\boldsymbol{x}^i_{\mathrm{mis}} = \int p_{\boldsymbol{\theta}}(\boldsymbol{x}^i_{\mathrm{obs}} \mid \boldsymbol{z}) p_{\boldsymbol{\theta}}(\boldsymbol{z})\, \mathrm{d}\boldsymbol{z}, \quad (1)$$

where the inner integral $\int p_{\boldsymbol{\theta}}(\boldsymbol{x}_{\mathrm{obs}}, \boldsymbol{x}_{\mathrm{mis}} \mid \boldsymbol{z})\, \mathrm{d}\boldsymbol{x}_{\mathrm{mis}}$ is often computationally tractable in VAEs due to standard assumptions, such as the conditional independence of $\boldsymbol{x}$ given $\boldsymbol{z}$ or the use of the Gaussian family for the decoder $p_{\boldsymbol{\theta}}(\boldsymbol{x} \mid \boldsymbol{z})$. Similar to existing work, we also make the assumption that the marginalisation of the missing variables is tractable. However, the marginal likelihood above remains intractable to compute as a consequence of the integral over the latents $\boldsymbol{z}$.

Due to the intractable integral, VAEs are typically fitted via a variational evidence lower-bound (ELBO)

$$\log p_{\boldsymbol{\theta}}(\boldsymbol{y}) \geq \mathbb{E}_{q_{\boldsymbol{\phi}}(\boldsymbol{z}\mid\boldsymbol{y})} \left[ \log \frac{p_{\boldsymbol{\theta}}(\boldsymbol{y} \mid \boldsymbol{z}) p_{\boldsymbol{\theta}}(\boldsymbol{z})}{q_{\boldsymbol{\phi}}(\boldsymbol{z} \mid \boldsymbol{y})} \right] = \log p_{\boldsymbol{\theta}}(\boldsymbol{y}) - D_{\mathrm{KL}}(q_{\boldsymbol{\phi}}(\boldsymbol{z} \mid \boldsymbol{y}) \,||\, p_{\boldsymbol{\theta}}(\boldsymbol{z} \mid \boldsymbol{y})), \quad (2)$$

where $\boldsymbol{y}$ refers to $\boldsymbol{x}^i$ in the fully-observed case, and to $\boldsymbol{x}^i_{\mathrm{obs}}$ in the incomplete-data case, and $q_{\boldsymbol{\phi}}(\boldsymbol{z} \mid \boldsymbol{y})$ is an (amortised) variational distribution with parameters $\boldsymbol{\phi}$ that is shared for all data-points in the data set (Gershman & Goodman, 2014). The amortised distribution is parametrised using a neural network (the encoder), which takes the data-point $\boldsymbol{y}$ as the input and predicts the distributional parameters of the variational family. Moreover, when the data is incomplete, i.e. $\boldsymbol{y} = \boldsymbol{x}^i_{\mathrm{obs}}$, sharing of the encoder for any pattern of missingness is often achieved by fixing the input dimensionality of the encoder to twice the size of $\boldsymbol{x}$ and providing $\gamma(\boldsymbol{x}^i_{\mathrm{obs}})$ and $\boldsymbol{m}^i$ as the inputs,[2] where $\gamma(\cdot)$ is a function that takes the incomplete data-point $\boldsymbol{x}_{\mathrm{obs}}$ and produces a vector of length $D$ with the missing dimensions set to zero[3] (Nazábal et al., 2020; Mattei & Frellsen, 2019).

From eq. (2), we see that the training objective for incomplete and fully-observed data has the same form, and therefore it may seem that fitting VAEs from incomplete data would be similarly difficult to the fully-observed case. However, as we will see next, data missingness can make model estimation much harder than in the complete data case.

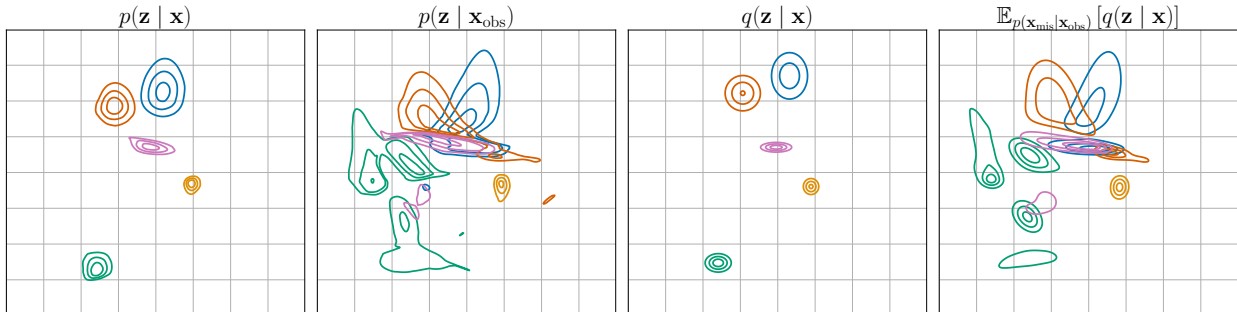

Figure 1: *Illustration of the posterior complexity due to missing data.* Each colour represents a different data-point $\boldsymbol{x}^i$. First: the model posterior $p_{\boldsymbol{\theta}}(\boldsymbol{z} \mid \boldsymbol{x})$ under complete data $\boldsymbol{x}$. Second: the model posterior $p_{\boldsymbol{\theta}}(\boldsymbol{z} \mid \boldsymbol{x}_{\text{obs}})$ under incomplete data $\boldsymbol{x}_{\text{obs}}$. Third: variational approximation $q_{\boldsymbol{\phi}}(\boldsymbol{z} \mid \boldsymbol{x})$ of the complete-data posterior $p_{\boldsymbol{\theta}}(\boldsymbol{z} \mid \boldsymbol{x})$. Fourth: an imputation-mixture variational approximation $\mathbb{E}_{p_{\boldsymbol{\theta}}(\boldsymbol{x}_{\text{mis}}|\boldsymbol{x}_{\text{obs}})}[q_{\boldsymbol{\phi}}(\boldsymbol{z} \mid \boldsymbol{x}_{\text{obs}}, \boldsymbol{x}_{\text{mis}})]$ of the incomplete posterior $p_{\boldsymbol{\theta}}(\boldsymbol{z} \mid \boldsymbol{x}_{\text{obs}})$. In these figures, we use a VAE with Gaussian variational, prior, and decoder distributions fitted on complete data, then the incomplete data-points $\boldsymbol{x}_{\text{obs}}$ are obtained by randomly masking 50% of the values from the complete data-points $\boldsymbol{x}$.

## 3 Implications of incomplete data for VAE estimation

The decomposition of the ELBO in eq. (2) emphasises that accurate estimation of the VAE model requires the variational distribution $q_{\boldsymbol{\phi}}(\boldsymbol{z} \mid \boldsymbol{x}_{\text{obs}})$ to accurately approximate the model posterior $p_{\boldsymbol{\theta}}(\boldsymbol{z} \mid \boldsymbol{x}_{\text{obs}})$. While it might appear that the marginalisation of the missing variables in eq. (1) comes at no cost since the ELBO maintains the same form as in the complete case, we here illustrate that his is not the case.

In the two left-most columns of fig. 1 we illustrate the model posteriors $p_{\boldsymbol{\theta}}(\boldsymbol{z} \mid \cdot)$ under fully-observed data $\boldsymbol{x}$ and partially-observed data $\boldsymbol{x}_{\text{obs}}$.[4] We discover that the model posteriors $p_{\boldsymbol{\theta}}(\boldsymbol{z} \mid \boldsymbol{x})$, which exhibited a certain regularity in the complete-data scenario, have become irregular multimodal distributions $p_{\boldsymbol{\theta}}(\boldsymbol{z} \mid \boldsymbol{x}_{\text{obs}})$ in the case of incomplete data.[5] Hence, accurate estimation of VAEs from incomplete data may require more flexible variational families than in the fully-observed case: while a family may sufficiently well approximate the model posterior in the fully-observed case, it may no longer be sufficiently flexible in the incomplete data case. We provide a further explanation when this situation may occur in appendix A. As a result of the mismatch between the model posterior $p_{\boldsymbol{\theta}}(\boldsymbol{z} \mid \boldsymbol{x}_{\text{obs}})$ and the variational distribution $q_{\boldsymbol{\phi}}(\boldsymbol{z} \mid \boldsymbol{x}_{\text{obs}})$, the incomplete-data KL divergence term $D_{\text{KL}}(q_{\boldsymbol{\phi}}(\boldsymbol{z} \mid \boldsymbol{x}_{\text{obs}}) \,\|\, p_{\boldsymbol{\theta}}(\boldsymbol{z} \mid \boldsymbol{x}_{\text{obs}}))$ in eq. (2) may be large compared to the analogous KL term $D_{\text{KL}}(q_{\boldsymbol{\phi}}(\boldsymbol{z} \mid \boldsymbol{x}) \,\|\, p_{\boldsymbol{\theta}}(\boldsymbol{z} \mid \boldsymbol{x}))$ in the complete-data case, subsequently introducing a bias to the fit of the model.

In the two right-most columns of fig. 1 we illustrate the variational approximations of the aforementioned model posterior distributions, $p_{\boldsymbol{\theta}}(\boldsymbol{z} \mid \boldsymbol{x})$ and $p_{\boldsymbol{\theta}}(\boldsymbol{z} \mid \boldsymbol{x}_{\text{obs}})$. The first of the two plots shows the complete-data variational distribution $q_{\boldsymbol{\phi}}(\boldsymbol{z} \mid \boldsymbol{x})$ obtained after training, which well-approximates the model posterior $p_{\boldsymbol{\theta}}(\boldsymbol{z} \mid \boldsymbol{x})$. In the second of the two plots, we construct the incomplete-data posterior approximation as follows: $p_{\boldsymbol{\theta}}(\boldsymbol{z} \mid \boldsymbol{x}_{\text{obs}}) = \mathbb{E}_{p_{\boldsymbol{\theta}}(\boldsymbol{x}_{\text{mis}}|\boldsymbol{x}_{\text{obs}})}[p_{\boldsymbol{\theta}}(\boldsymbol{z} \mid \boldsymbol{x}_{\text{obs}}, \boldsymbol{x}_{\text{mis}})] \approx \mathbb{E}_{p_{\boldsymbol{\theta}}(\boldsymbol{x}_{\text{mis}}|\boldsymbol{x}_{\text{obs}})}[q_{\boldsymbol{\phi}}(\boldsymbol{z} \mid \boldsymbol{x}_{\text{obs}}, \boldsymbol{x}_{\text{mis}})]$, which well-approximates the model posterior $p_{\boldsymbol{\theta}}(\boldsymbol{z} \mid \boldsymbol{x}_{\text{obs}})$ too. Taken together, the two plots show that if the variational family used in the fully-observed case well-approximates the model posterior, i.e. $q_{\boldsymbol{\phi}}(\boldsymbol{z} \mid \boldsymbol{x}) \approx p_{\boldsymbol{\theta}}(\boldsymbol{z} \mid \boldsymbol{x})$, then the imputation-mixture $\mathbb{E}_{p_{\boldsymbol{\theta}}(\boldsymbol{x}_{\text{mis}}|\boldsymbol{x}_{\text{obs}})}[q_{\boldsymbol{\phi}}(\boldsymbol{z} \mid \boldsymbol{x}_{\text{obs}}, \boldsymbol{x}_{\text{mis}})]$ will also be a good approximation of the incomplete-data posterior $p_{\boldsymbol{\theta}}(\boldsymbol{z} \mid \boldsymbol{x}_{\text{obs}})$. This observation suggests that we can work with the same variational family in both the fully-observed and incomplete data scenarios if we adopt a mixture approach. In the rest of

---

[1]While there is some historical disparity between MAR assumptions in the statistics literature, we can here adopt the weakest MAR assumption, also known as the *realised* MAR (Seaman et al., 2013).

[2]Alternative encoder architectures, such as, permutation-invariant networks (Ma et al., 2019) are also used.

[3]Equivalent to setting the missing dimensions to the empirical mean for zero-centered data.

[4]In fig. 1 we use a VAE with Gaussian variational, prior, and decoder distributions fitted on complete data.

[5]A related phenomenon, called posterior inconsistency, has been recently reported in concurrent work by Sudak & Tschiatschek (2023), relating $p_{\boldsymbol{\theta}}(\boldsymbol{z} \mid \boldsymbol{x}_{\text{obs}})$ and $p_{\boldsymbol{\theta}}(\boldsymbol{z} \mid \boldsymbol{x}_{\text{obs}\backslash u})$, where $u$ is a subset of the observed dimensions (see section 5).

this paper, we investigate opportunities to improve VAE estimation from incomplete data by constructing variational mixture approximations of the incomplete-data posterior.

## 4  Fitting VAEs from incomplete data using mixture variational families

We propose working with mixture variational families to mitigate the increase in posterior complexity and to improve the estimation accuracy of VAEs when the training data are incomplete. This allows us to use families of distributions for the mixture components that are known to work well when the data is fully-observed, and use the mixtures to handle the increased posterior complexity due to missing data.

We propose two approaches for constructing variational mixtures. In section 4.1 we specify $q_{\boldsymbol{\phi}}(\boldsymbol{z} \mid \boldsymbol{x}_{\mathrm{obs}})$ as a finite-mixture distribution that can be learnt directly using the reparametrisation trick. In section 4.2 we investigate an imputation-based variational-mixture to approximate $\mathbb{E}_{p_{\boldsymbol{\theta}}(\boldsymbol{x}_{\mathrm{mis}}|\boldsymbol{x}_{\mathrm{obs}})}[q_{\boldsymbol{\phi}}(\boldsymbol{z} \mid \boldsymbol{x}_{\mathrm{obs}}, \boldsymbol{x}_{\mathrm{mis}})]$. Detailed evaluation of the proposed methods is provided in section 6.

### 4.1  Using finite mixture variational distributions to fit VAEs from incomplete data

In section 3 we saw that the imputation-mixture $\mathbb{E}_{p_{\boldsymbol{\theta}}(\boldsymbol{x}_{\mathrm{mis}}|\boldsymbol{x}_{\mathrm{obs}})}[q_{\boldsymbol{\phi}}(\boldsymbol{z} \mid \boldsymbol{x}_{\mathrm{obs}}, \boldsymbol{x}_{\mathrm{mis}})]$ is a good approximation of the incomplete-data posterior $p_{\boldsymbol{\theta}}(\boldsymbol{z} \mid \boldsymbol{x}_{\mathrm{obs}})$ and would thus be a suitable variational distribution $q_{\boldsymbol{\phi}}(\boldsymbol{z} \mid \boldsymbol{x}_{\mathrm{obs}})$. However, estimation of the imputation distribution $p_{\boldsymbol{\theta}}(\boldsymbol{x}_{\mathrm{mis}} \mid \boldsymbol{x}_{\mathrm{obs}})$ is generally intractable for VAEs (Rezende et al., 2014; Mattei & Frellsen, 2018a; Simkus & Gutmann, 2023). Hence, we here consider a more tractable approach and specify the variational distribution $q_{\boldsymbol{\phi}}(\boldsymbol{z} \mid \boldsymbol{x}_{\mathrm{obs}})$ in terms of a finite-mixture distribution:

$$q_{\boldsymbol{\phi}}(\boldsymbol{z} \mid \boldsymbol{x}_{\mathrm{obs}}) = \sum_{k=1}^{K} q_{\boldsymbol{\phi}}(k \mid \boldsymbol{x}_{\mathrm{obs}}) q_{\boldsymbol{\phi}}^{k}(\boldsymbol{z} \mid \boldsymbol{x}_{\mathrm{obs}}), \tag{3}$$

where $q_{\boldsymbol{\phi}}(k \mid \boldsymbol{x}_{\mathrm{obs}})$ is a categorical distribution over the components $k \in \{1, \ldots, K\}$ and each component distribution $q_{\boldsymbol{\phi}}^{k}(\boldsymbol{z} \mid \boldsymbol{x}_{\mathrm{obs}})$ belongs to any reparametrisable distribution family. Both $q_{\boldsymbol{\phi}}(k \mid \boldsymbol{x}_{\mathrm{obs}})$ and $q_{\boldsymbol{\phi}}^{k}(\boldsymbol{z} \mid \boldsymbol{x}_{\mathrm{obs}})$ are amortised using an encoder network, similar to section 2.

The "reparametrisation trick" is typically used in VAEs to efficiently optimise the parameters $\boldsymbol{\phi}$ of the variational distribution $q_{\boldsymbol{\phi}}(\boldsymbol{z} \mid \boldsymbol{x}_{\mathrm{obs}})$. This requires that the random variable $\boldsymbol{z}$ can be parametrised as a learnable differentiable transformation $t(\boldsymbol{\epsilon}; \boldsymbol{x}_{\mathrm{obs}}, \boldsymbol{\phi})$ of another random variable $\boldsymbol{\epsilon}$ that follows a distribution with no learnable parameters. However, reparametrising mixture-families requires extra care: sampling the mixture $q_{\boldsymbol{\phi}}(\boldsymbol{z} \mid \boldsymbol{x}_{\mathrm{obs}})$ in eq. (3) is typically done via ancestral sampling by first drawing $k \sim q_{\boldsymbol{\phi}}(k \mid \boldsymbol{x}_{\mathrm{obs}})$ and then $\boldsymbol{z} \sim q_{\boldsymbol{\phi}}^{k}(\boldsymbol{z} \mid \boldsymbol{x}_{\mathrm{obs}})$, but the sampling of the categorical distribution $q_{\boldsymbol{\phi}}(k \mid \boldsymbol{x}_{\mathrm{obs}})$ is non-differentiable, making the direct application of the "reparametrisation trick" generally infeasible.

To make the fitting of VAEs using mixture-variational distributions feasible we consider two objectives based on the variational ELBO (Kingma & Welling, 2013; Rezende et al., 2014):

$$\mathcal{L}_{\mathrm{ELBO}}(\boldsymbol{x}_{\mathrm{obs}}) = \mathbb{E}_{q_{\boldsymbol{\phi}}(\boldsymbol{z}|\boldsymbol{x}_{\mathrm{obs}})} \left[\log w(\boldsymbol{z})\right], \quad \text{and} \tag{4}$$

$$\mathcal{L}_{\mathrm{SELBO}}(\boldsymbol{x}_{\mathrm{obs}}) = \sum_{k=1}^{K} q_{\boldsymbol{\phi}}(k \mid \boldsymbol{x}_{\mathrm{obs}}) \mathbb{E}_{q_{\boldsymbol{\phi}}^{k}(\boldsymbol{z}|\boldsymbol{x}_{\mathrm{obs}})} \left[\log w(\boldsymbol{z})\right], \tag{5}$$

$$\text{where} \quad w(\boldsymbol{z}) = \frac{p_{\boldsymbol{\theta}}(\boldsymbol{x}_{\mathrm{obs}}, \boldsymbol{z})}{q_{\boldsymbol{\phi}}(\boldsymbol{z} \mid \boldsymbol{x}_{\mathrm{obs}})}. \tag{6}$$

The first objective $\mathcal{L}_{\mathrm{ELBO}}$ corresponds to the standard ELBO, while $\mathcal{L}_{\mathrm{SELBO}}$ is the stratified ELBO (Roeder et al., 2017, Section 4; Morningstar et al., 2021). When working with $\mathcal{L}_{\mathrm{ELBO}}$, due to the mixture variational family, we will need to optimise $\boldsymbol{\phi}$ with *implicit* reparametrisation (Figurnov et al., 2019). Implicit reparametrisation of mixture distributions requires that the component distributions $q_{\boldsymbol{\phi}}^{k}(\boldsymbol{z} \mid \boldsymbol{x}_{\mathrm{obs}})$ can be factorised using the chain rule, i.e. $q_{\boldsymbol{\phi}}^{k}(\boldsymbol{z} \mid \boldsymbol{x}_{\mathrm{obs}}) = \prod_d q_{\boldsymbol{\phi}}^{k}(z_d \mid \boldsymbol{z}_{<d}, \boldsymbol{x}_{\mathrm{obs}})$, and that we have access to the CDF (or other standardisation function) of each factor $q_{\boldsymbol{\phi}}^{k}(z_d \mid \boldsymbol{z}_{<d}, \boldsymbol{x}_{\mathrm{obs}})$. However, the chain rule requirement can be difficult to satisfy for some highly flexible variational families, such as normalising flows (e.g.

Papamakarios et al., 2021), and finding the (conditional) CDF of the factors can also be hard if not already known in closed form. Consequently, $\mathcal{L}_{\mathrm{ELBO}}$ with implicit reparametrisation may not be usable with all distribution families as components of the variational mixture.

The second objective $\mathcal{L}_{\mathrm{SELBO}}$, on the other hand, samples the mixture distribution with stratified sampling,[6] which avoids the non-differentiability of sampling $q_{\boldsymbol{\phi}}(k \mid \boldsymbol{x}_{\mathrm{obs}})$, and as a result allows us to use any family of reparametrisable distributions as the mixture components.

The importance-weighted ELBO (IWELBO, Burda et al., 2015) is often used as an alternative to the standard ELBO as it can be made tighter. We here also consider an ordinary version, $\mathcal{L}_{\mathrm{IWELBO}}$, and a stratified version, $\mathcal{L}_{\mathrm{SIWELBO}}$ (Shi et al., 2019, Appendix A; Morningstar et al., 2021):

$$\mathcal{L}_{\mathrm{IWELBO}}^{I}(\boldsymbol{x}_{\mathrm{obs}}) = \mathbb{E}_{\{\boldsymbol{z}_j\}_{j=1}^{I} \sim q_{\boldsymbol{\phi}}(\boldsymbol{z}|\boldsymbol{x}_{\mathrm{obs}})}\left[\log \frac{1}{I}\sum_{j=1}^{I} w(\boldsymbol{z}_j)\right], \quad \text{and} \tag{7}$$

$$\mathcal{L}_{\mathrm{SIWELBO}}^{I}(\boldsymbol{x}_{\mathrm{obs}}) = \mathbb{E}_{\{\{\boldsymbol{z}_j^k\}_{j=1}^{I} \sim q_{\boldsymbol{\phi}}^k(\boldsymbol{z}|\boldsymbol{x}_{\mathrm{obs}})\}_{k=1}^{K}}\left[\log \sum_{k=1}^{K} q_{\boldsymbol{\phi}}(k \mid \boldsymbol{x}_{\mathrm{obs}})\frac{1}{I}\sum_{j=1}^{I} w(\boldsymbol{z}_j^k)\right],[7] \tag{8}$$

where $I$ is the number of importance samples in $\mathcal{L}_{\mathrm{IWELBO}}$ and the number of samples per-mixture-component in $\mathcal{L}_{\mathrm{SIWELBO}}$.

When the number of mixture-components is $K = 1$ the lower-bounds in eqs. (4-5) and eqs. (7-8) correspond to the MVAE and MIWAE bounds in Mattei & Frellsen (2019) which are among the most popular bounds for fitting VAEs from incomplete data. However, as $K > 1$ the proposed bounds can be tighter due to an increased flexibility of the variational distribution $q_{\boldsymbol{\phi}}(\boldsymbol{z} \mid \boldsymbol{x}_{\mathrm{obs}})$ (Morningstar et al., 2021, Appendix A), which potentially mitigates the problems caused by the missing data (see section 3). Finally, the importance-weighted bounds in eqs. (7) and (8) maintain the asymptotic consistency guarantees of Burda et al. (2015) and approaches the true marginal log-likelihood $\log p_{\boldsymbol{\theta}}(\boldsymbol{x}_{\mathrm{obs}})$ as $K \cdot I \to \infty$, allowing for more accurate estimation of the model with increasing computational budget.

We denote the four methods based on eqs. (4), (5), (7) and (8) by **MissVAE**, **MissSVAE**, **MissIWAE**, and **MissSIWAE** respectively.

## 4.2 Using imputation-mixture distributions to fit VAEs from incomplete data

In section 4.1, we jointly dealt with both the inference of the latents $\boldsymbol{z}$ (section 2) and the posterior complexity increase due to missing data (section 3) by learning a finite-mixture variational distribution. Here, we propose a second "decomposed" approach to deal with the complexities of missing data.

Intuitively, if we had an oracle that were able to generate imputations of the missing data from the ground truth conditional distribution $p^*(\boldsymbol{x}_{\mathrm{mis}} \mid \boldsymbol{x}_{\mathrm{obs}})$, then the VAE estimation task would reduce to the case of complete-data. This suggests that an *effective strategy is to decompose the task of model estimation from incomplete data into two (iterative) tasks: (i) data imputation and (ii) model estimation*, akin to the Monte Carlo EM algorithm (Wei & Tanner, 1990; Dempster et al., 1977). However, access to the oracle $p^*(\boldsymbol{x}_{\mathrm{mis}} \mid \boldsymbol{x}_{\mathrm{obs}})$ is unrealistic and the exact sampling of $p_{\boldsymbol{\theta}}(\boldsymbol{x}_{\mathrm{mis}} \mid \boldsymbol{x}_{\mathrm{obs}})$, as required in EM, is generally intractable. To address this, we resort to (i) approximate but computationally cheap conditional sampling methods for VAEs to generate imputations (Rezende et al., 2014; Mattei & Frellsen, 2018a; Simkus &

---

[6]Stratified sampling of mixture distributions typically draws an equal number of samples from each component and weighs the samples by the component probabilities $q_{\boldsymbol{\phi}}(k \mid \boldsymbol{x}_{\mathrm{obs}})$ when estimating expectations. It is commonly used to reduce Monte Carlo variance (Robert & Casella, 2004).

[7]In multimodal-domain VAE literature, Shi et al. (2019) proposed a looser bound related to $\mathcal{L}_{\mathrm{SIWELBO}}$:

$$\mathcal{L}_{\mathrm{SIWELBO}}^{I}(\boldsymbol{x}_{\mathrm{obs}}) \geq \tilde{\mathcal{L}}_{\mathrm{SIWELBO}}^{I}(\boldsymbol{x}_{\mathrm{obs}}) \overset{\text{def}}{=} \sum_{k=1}^{K} q_{\boldsymbol{\phi}}(k \mid \boldsymbol{x}_{\mathrm{obs}})\mathbb{E}_{\{\boldsymbol{z}_j^k\}_{j=1}^{I} \sim q_{\boldsymbol{\phi}}^k(\boldsymbol{z}|\boldsymbol{x}_{\mathrm{obs}})}\left[\log \frac{1}{I}\sum_{j=1}^{I} w(\boldsymbol{z}_j^k)\right] \overset{I=1}{=} \mathcal{L}_{\mathrm{SELBO}}(\boldsymbol{x}_{\mathrm{obs}}),$$

and empirically showed that it may alleviate potential mixture collapse to a subset of the mixture components. Therefore, the looser bound may be useful when variational mixture collapse is observed.

Gutmann, 2023) and (ii) separate learning objectives for the model $p_{\boldsymbol{\theta}}$ and the variational distribution $q_{\boldsymbol{\phi}}$ to compensate for potential sampling errors. We call the proposed approach **DeMissVAE** (**de**composed approach for handling **miss**ing data in **VAE**s).

We construct the variational distribution $q_{\boldsymbol{\phi}, f^t}(\boldsymbol{z} \mid \boldsymbol{x}_{\mathrm{obs}})$ for an incomplete data-point $\boldsymbol{x}_{\mathrm{obs}}$ using a completed-data variational distribution $q_{\boldsymbol{\phi}}(\boldsymbol{z} \mid \boldsymbol{x}_{\mathrm{obs}}, \boldsymbol{x}_{\mathrm{mis}})$ and an (approximate) imputation distribution $f^t(\boldsymbol{x}_{\mathrm{mis}} \mid \boldsymbol{x}_{\mathrm{obs}}) \approx p_{\boldsymbol{\theta}}(\boldsymbol{x}_{\mathrm{mis}} \mid \boldsymbol{x}_{\mathrm{obs}})$:

$$q_{\boldsymbol{\phi}, f^t}(\boldsymbol{z} \mid \boldsymbol{x}_{\mathrm{obs}}) = \mathbb{E}_{f^t(\boldsymbol{x}_{\mathrm{mis}} \mid \boldsymbol{x}_{\mathrm{obs}})} \left[ q_{\boldsymbol{\phi}}(\boldsymbol{z} \mid \boldsymbol{x}_{\mathrm{obs}}, \boldsymbol{x}_{\mathrm{mis}}) \right]. \tag{9}$$

The intuition for this construction comes from the decomposition of the model posterior $p_{\boldsymbol{\theta}}(\boldsymbol{z} \mid \boldsymbol{x}_{\mathrm{obs}}) = \mathbb{E}_{p_{\boldsymbol{\theta}}(\boldsymbol{x}_{\mathrm{mis}} \mid \boldsymbol{x}_{\mathrm{obs}})}[p_{\boldsymbol{\theta}}(\boldsymbol{z} \mid \boldsymbol{x}_{\mathrm{obs}}, \boldsymbol{x}_{\mathrm{mis}})]$. Assuming that the completed-data variational distribution $q_{\boldsymbol{\phi}}(\boldsymbol{z} \mid \boldsymbol{x}_{\mathrm{obs}}, \boldsymbol{x}_{\mathrm{mis}})$ well-represents the model posterior $p_{\boldsymbol{\theta}}(\boldsymbol{z} \mid \boldsymbol{x}_{\mathrm{obs}}, \boldsymbol{x}_{\mathrm{mis}})$, and that the imputation distribution $f^t(\boldsymbol{x}_{\mathrm{mis}} \mid \boldsymbol{x}_{\mathrm{obs}})$ draws plausible imputations of the missing variables, then $q_{\boldsymbol{\phi}, f^t}(\boldsymbol{z} \mid \boldsymbol{x}_{\mathrm{obs}})$ will reasonably represent $p_{\boldsymbol{\theta}}(\boldsymbol{z} \mid \boldsymbol{x}_{\mathrm{obs}})$ (see the two right-most columns of fig. 1). A more technical justification is provided below, in the paragraph after eq. (13). In contrast to section 4.1 we here use a continuous-mixture variational distribution, which is more flexible than a finite-mixture distribution, albeit at an extra computational cost due to sampling the (approximate) imputations (see appendix D).

We now derive the DeMissVAE objectives for fitting the generative model $p_{\boldsymbol{\theta}}(\boldsymbol{x})$ and the completed-data variational distribution $q_{\boldsymbol{\phi}}(\boldsymbol{z} \mid \boldsymbol{x}_{\mathrm{obs}}, \boldsymbol{x}_{\mathrm{mis}})$, see appendix C for a more in-depth treatment.

**Objective for $p_{\boldsymbol{\theta}}(\boldsymbol{x}, \boldsymbol{z})$.** With the variational distribution in eq. (9), we derive an ELBO on the marginal log-likelihood, similar to eq. (2), to learn the parameters $\boldsymbol{\theta}$ of the generative model:

$$\log p_{\boldsymbol{\theta}}(\boldsymbol{x}_{\mathrm{obs}}) \geq \mathbb{E}_{f^t(\boldsymbol{x}_{\mathrm{mis}} \mid \boldsymbol{x}_{\mathrm{obs}}) q_{\boldsymbol{\phi}}(\boldsymbol{z} \mid \boldsymbol{x}_{\mathrm{obs}}, \boldsymbol{x}_{\mathrm{mis}})} \left[ \log \frac{p_{\boldsymbol{\theta}}(\boldsymbol{x}_{\mathrm{obs}}, \boldsymbol{z})}{\mathbb{E}_{f^t(\boldsymbol{x}_{\mathrm{mis}} \mid \boldsymbol{x}_{\mathrm{obs}})} \left[ q_{\boldsymbol{\phi}}(\boldsymbol{z} \mid \boldsymbol{x}_{\mathrm{obs}}, \boldsymbol{x}_{\mathrm{mis}}) \right]} \right]$$

$$= \underbrace{\mathbb{E}_{f^t(\boldsymbol{x}_{\mathrm{mis}} \mid \boldsymbol{x}_{\mathrm{obs}}) q_{\boldsymbol{\phi}}(\boldsymbol{z} \mid \boldsymbol{x}_{\mathrm{obs}}, \boldsymbol{x}_{\mathrm{mis}})} \left[ \log p_{\boldsymbol{\theta}}(\boldsymbol{x}_{\mathrm{obs}}, \boldsymbol{z}) \right]}_{\overset{\mathrm{def}}{=} \mathcal{L}_{\mathrm{CVI}}^{\boldsymbol{\theta}}(\boldsymbol{x}_{\mathrm{obs}}; \boldsymbol{\phi}, \boldsymbol{\theta}, f^t)} + \underbrace{\mathcal{H} \left[ q_{\boldsymbol{\phi}, f^t}(\boldsymbol{z} \mid \boldsymbol{x}_{\mathrm{obs}}) \right]}_{\text{Const. w.r.t. } \boldsymbol{\theta}}. \tag{10}$$

This lower-bound can be further decomposed into log-likelihood and KL divergence terms

$$\mathcal{L}_{\mathrm{CVI}}^{\boldsymbol{\theta}}(\boldsymbol{x}_{\mathrm{obs}}; \boldsymbol{\phi}, \boldsymbol{\theta}, f^t) + \mathcal{H} \left[ q_{\boldsymbol{\phi}, f^t}(\boldsymbol{z} \mid \boldsymbol{x}_{\mathrm{obs}}) \right] = \log p_{\boldsymbol{\theta}}(\boldsymbol{x}_{\mathrm{obs}}) - D_{\mathrm{KL}}(q_{\boldsymbol{\phi}, f^t}(\boldsymbol{z} \mid \boldsymbol{x}_{\mathrm{obs}}) \,\|\, p_{\boldsymbol{\theta}}(\boldsymbol{z} \mid \boldsymbol{x}_{\mathrm{obs}})), \tag{11}$$

which means that if $q_{\boldsymbol{\phi}, f^t}(\boldsymbol{z} \mid \boldsymbol{x}_{\mathrm{obs}}) \approx p_{\boldsymbol{\theta}}(\boldsymbol{z} \mid \boldsymbol{x}_{\mathrm{obs}})$ then maximising eq. (10) w.r.t. $\boldsymbol{\theta}$ performs approximate maximum-likelihood estimation. Importantly, the missing variables $\boldsymbol{x}_{\mathrm{mis}}$ are marginalised-out, which adds robustness to the potential sampling errors in $f^t(\boldsymbol{x}_{\mathrm{mis}} \mid \boldsymbol{x}_{\mathrm{obs}})$.

**Objective for $q_{\boldsymbol{\phi}}(\boldsymbol{z} \mid \boldsymbol{x})$.** We obtain the objective for learning the variational distribution $q_{\boldsymbol{\phi}}(\boldsymbol{z} \mid \boldsymbol{x})$ by marginalising the missing variables $\boldsymbol{x}_{\mathrm{mis}}$ from the complete-data ELBO in eq. (2) and then lower-bounding the integral using $f^t(\boldsymbol{x}_{\mathrm{mis}} \mid \boldsymbol{x}_{\mathrm{obs}})$ (see appendix B):

$$\log p_{\boldsymbol{\theta}}(\boldsymbol{x}_{\mathrm{obs}}) \geq \underbrace{\mathbb{E}_{f^t(\boldsymbol{x}_{\mathrm{mis}} \mid \boldsymbol{x}_{\mathrm{obs}}) q_{\boldsymbol{\phi}}(\boldsymbol{z} \mid \boldsymbol{x}_{\mathrm{obs}}, \boldsymbol{x}_{\mathrm{mis}})} \left[ \log \frac{p_{\boldsymbol{\theta}}(\boldsymbol{x}_{\mathrm{obs}}, \boldsymbol{x}_{\mathrm{mis}}, \boldsymbol{z})}{q_{\boldsymbol{\phi}}(\boldsymbol{z} \mid \boldsymbol{x}_{\mathrm{obs}}, \boldsymbol{x}_{\mathrm{mis}})} \right]}_{\overset{\mathrm{def}}{=} \mathcal{L}_{\mathrm{LMVB}}^{\boldsymbol{\phi}}(\boldsymbol{x}_{\mathrm{obs}}; \boldsymbol{\phi}, \boldsymbol{\theta}, f^t)} + \underbrace{\mathcal{H} \left[ f^t(\boldsymbol{x}_{\mathrm{mis}} \mid \boldsymbol{x}_{\mathrm{obs}}) \right]}_{\text{Const. w.r.t. } \boldsymbol{\phi}}. \tag{12}$$

This lower-bound can also be decomposed into the log-likelihood term and two KL divergence terms

$$\mathcal{L}_{\mathrm{LMVB}}^{\boldsymbol{\phi}}(\boldsymbol{x}_{\mathrm{obs}}; \boldsymbol{\phi}, \boldsymbol{\theta}, f^t) + \mathcal{H} \left[ f^t(\boldsymbol{x}_{\mathrm{mis}} \mid \boldsymbol{x}_{\mathrm{obs}}) \right] = \log p_{\boldsymbol{\theta}}(\boldsymbol{x}_{\mathrm{obs}}) - D_{\mathrm{KL}}(f^t(\boldsymbol{x}_{\mathrm{mis}} \mid \boldsymbol{x}_{\mathrm{obs}}) \,\|\, p_{\boldsymbol{\theta}}(\boldsymbol{x}_{\mathrm{mis}} \mid \boldsymbol{x}_{\mathrm{obs}}))$$
$$- \mathbb{E}_{f^t(\boldsymbol{x}_{\mathrm{mis}} \mid \boldsymbol{x}_{\mathrm{obs}})} \left[ D_{\mathrm{KL}}(q_{\boldsymbol{\phi}}(\boldsymbol{z} \mid \boldsymbol{x}_{\mathrm{obs}}, \boldsymbol{x}_{\mathrm{mis}}) \,\|\, p_{\boldsymbol{\theta}}(\boldsymbol{z} \mid \boldsymbol{x}_{\mathrm{obs}}, \boldsymbol{x}_{\mathrm{mis}})) \right], \tag{13}$$

which means that the bound is maximised w.r.t. $\boldsymbol{\phi}$ iff $q_{\boldsymbol{\phi}}(\boldsymbol{z} \mid \boldsymbol{x}_{\mathrm{obs}}, \boldsymbol{x}_{\mathrm{mis}}) = p_{\boldsymbol{\theta}}(\boldsymbol{z} \mid \boldsymbol{x}_{\mathrm{obs}}, \boldsymbol{x}_{\mathrm{mis}})$ for all $\boldsymbol{x}_{\mathrm{mis}}$. Therefore, using the above objective to fit $q_{\boldsymbol{\phi}}$ corresponds directly to the complete-data case, and hence avoids having to approximate complex posteriors that arise due to missing data (see 3).

If $q_{\boldsymbol{\phi}}(\boldsymbol{z} \mid \boldsymbol{x}_{\text{obs}}, \boldsymbol{x}_{\text{mis}}) = p_{\boldsymbol{\theta}}(\boldsymbol{z} \mid \boldsymbol{x}_{\text{obs}}, \boldsymbol{x}_{\text{mis}})$ for all $\boldsymbol{x}_{\text{mis}}$, then maximising either of the bounds in eqs. (10) or (12) w.r.t. the imputation distribution $f^t(\boldsymbol{x}_{\text{mis}} \mid \boldsymbol{x}_{\text{obs}})$ would correspond to setting $f^t(\boldsymbol{x}_{\text{mis}} \mid \boldsymbol{x}_{\text{obs}}) = p_{\boldsymbol{\theta}}(\boldsymbol{x}_{\text{mis}} \mid \boldsymbol{x}_{\text{obs}})$. However, directly learning an imputation distribution $f^t(\boldsymbol{x}_{\text{mis}} \mid \boldsymbol{x}_{\text{obs}}) \approx p_{\boldsymbol{\theta}}(\boldsymbol{x}_{\text{mis}} \mid \boldsymbol{x}_{\text{obs}})$ is challenging (Simkus et al., 2023, Section 2.2). This motivates using sampling methods to approximate the optimal imputation distribution $f^t(\boldsymbol{x}_{\text{mis}} \mid \boldsymbol{x}_{\text{obs}}) \approx p_{\boldsymbol{\theta}}(\boldsymbol{x}_{\text{mis}} \mid \boldsymbol{x}_{\text{obs}})$ with samples. We draw samples from $f^t(\boldsymbol{x}_{\text{mis}} \mid \boldsymbol{x}_{\text{obs}})$ using (cheap) approximate conditional sampling methods for VAEs to obtain $K$ imputations $\{\boldsymbol{x}_{\text{mis}}^k\}_k^K$ and then use them to approximate the expectations w.r.t. $f^t(\boldsymbol{x}_{\text{mis}} \mid \boldsymbol{x}_{\text{obs}})$ in the above objectives. We discuss the implementation of the algorithm in detail in appendix D.

Finally, we note that the $\mathcal{L}_{\text{CVI}}^{\boldsymbol{\theta}}$ and $\mathcal{L}_{\text{LMVB}}^{\boldsymbol{\phi}}$ objectives in eqs. (10) and (12) are based on the standard ELBO. Extensions to the importance-weighted ELBO might improve the method further by increasing the flexibility of the variational posterior. However, unlike the standard ELBO used in eq. (10) where the density of the imputation-based variational-mixture $q_{\boldsymbol{\phi}, f^t}(\boldsymbol{z} \mid \boldsymbol{x}_{\text{obs}})$ can be dropped, IWELBO requires computing the density of the proposal distribution $q_{\boldsymbol{\phi}, f^t}(\boldsymbol{z} \mid \boldsymbol{x}_{\text{obs}})$, which is generally intractable. We hence leave this direction for future work.

## 5 Related work

**Fitting VAEs from incomplete data.** Since the seminal works of Kingma & Welling (2013) and Rezende et al. (2014), VAEs have been widely used for density estimation from incomplete data and various downstream tasks, primarily due to the computationally efficient marginalisation of the model in eq. (1). Vedantam et al. (2017) and Wu & Goodman (2018) explored the use of product-of-experts variational distributions, drawing inspiration from findings in the factor analysis case with incomplete data (Williams et al., 2018). Mattei & Frellsen (2019) used the importance-weighted ELBO (Burda et al., 2015) for training VAEs on incomplete training data sets. Ma et al. (2019) proposed the use of permutation invariant neural networks to parametrise the encoder network instead of relying on zero-masking. Nazábal et al. (2020) introduced hierarchical priors to handle incomplete heterogeneous training data. Simkus et al. (2023) proposed a general-purpose approach that is applicable to VAEs, not requiring the decoder distribution to be easily marginalisable. Here, we further develop the understanding of VAEs in the presence missing values in the training data set, and propose variational-mixtures as a natural approach to improve VAE estimation from incomplete data, building upon the motivation from imputation-mixtures discussed in section 3.

**Variational mixture distributions.** Mixture distributions have found widespread application in variational inference and VAE literature. Roeder et al. (2017) introduced the stratified ELBO corresponding to eq. (5). In the context of VAEs in multimodal domains, Shi et al. (2019, Appendix A) introduced the stratified IWELBO corresponding to eq. (8), but opted to use a looser bound instead, see footnote 7. These bounds were subsequently rediscovered by Morningstar et al. (2021) and Kviman et al. (2023), who investigated their use for VAE estimation in fully-observed data scenarios. Figurnov et al. (2019) introduced implicit reparametrisation, enabling gradient estimation for ancestrally-sampled mixtures, allowing the estimation of variational mixtures using eqs. (4) and (7). Here, we build on this prior work, asserting that variational-mixtures are well-suited for handling the posterior complexity increase due to missing data (see section 3). Moreover, the imputation-mixture distribution used in DeMissVAE is a novel type of variational mixtures specifically designed for incomplete data scenarios.

**Posterior complexity increase due to missing data.** Concurrent to this study, Sudak & Tschiatschek (2023) have brought attention to a phenomenon related to the increase in posterior complexity due to incomplete data, discussed in section 3. They noted that, for any $\boldsymbol{x}_{\text{obs}}$ and $\boldsymbol{x}_{\text{obs}\setminus u}$, where $u$ is a subset of the observed dimensions, the model posteriors $p_{\boldsymbol{\theta}}(\boldsymbol{z} \mid \boldsymbol{x}_{\text{obs}})$ and $p_{\boldsymbol{\theta}}(\boldsymbol{z} \mid \boldsymbol{x}_{\text{obs}\setminus u})$ should exhibit a strong dependency. However, because of the approximations in the variational posterior (see e.g. Cremer et al., 2018; Zhang et al., 2021), the variational approximations $q_{\boldsymbol{\phi}}(\boldsymbol{z} \mid \boldsymbol{x}_{\text{obs}})$ and $q_{\boldsymbol{\phi}}(\boldsymbol{z} \mid \boldsymbol{x}_{\text{obs}\setminus u})$ may not consistently capture this dependency. They refer to the lack of dependency between $q_{\boldsymbol{\phi}}(\boldsymbol{z} \mid \boldsymbol{x}_{\text{obs}})$ and $q_{\boldsymbol{\phi}}(\boldsymbol{z} \mid \boldsymbol{x}_{\text{obs}\setminus u})$, compared to $p_{\boldsymbol{\theta}}(\boldsymbol{z} \mid \boldsymbol{x}_{\text{obs}})$ and $p_{\boldsymbol{\theta}}(\boldsymbol{z} \mid \boldsymbol{x}_{\text{obs}\setminus u})$, as posterior inconsistency. Focused on improving downstream task performance, they introduce regularisation into the VAE training objective to address posterior inconsistency. In contrast to their work, we compare the fully-observed and incomplete-data posteriors, $p_{\boldsymbol{\theta}}(\boldsymbol{z} \mid \boldsymbol{x})$ and

| Method | $p_{\boldsymbol{\theta}}$ objective | $q_{\boldsymbol{\phi}}$ objective | # of components | Mixture sampling |
|---|---|---|---|---|
| MVAE[†] | eq. (4) | eq. (4) | $K = 1$ | — |
| MissVAE | eq. (4) | eq. (4) | $K > 1$ | Ancestral |
| MissSVAE | eq. (5) | eq. (5) | $K > 1$ | Stratified |
| MIWAE[†] | eq. (7) | eq. (7) | $K = 1$ | — |
| MissIWAE | eq. (7) | eq. (7) | $K > 1$ | Ancestral |
| MissSIWAE | eq. (8) | eq. (8) | $K > 1$ | Stratified |
| DeMissVAE | eq. (10) | eq. (12) | $K > 1$ | Conditional VAE |

Table 1: *Summary of the proposed and baseline methods.* The non-mixture baselines (†) are based on Mattei & Frellsen (2019) and the other methods are proposed in this paper. Moreover, the methods using ancestral sampling require implicit reparametrisation (Figurnov et al., 2019), whereas the other methods work with the standard reparametrisation trick.

$p_{\boldsymbol{\theta}}(\boldsymbol{z} \mid \boldsymbol{x}_{\mathrm{obs}})$, respectively. Observing that the incomplete-data posterior $p_{\boldsymbol{\theta}}(\boldsymbol{z} \mid \boldsymbol{x}_{\mathrm{obs}})$ can be expressed as a mixture of fully-observed posteriors $p_{\boldsymbol{\theta}}(\boldsymbol{z} \mid \boldsymbol{x})$, that is, $p_{\boldsymbol{\theta}}(\boldsymbol{z} \mid \boldsymbol{x}_{\mathrm{obs}}) = \mathbb{E}_{p_{\boldsymbol{\theta}}(\boldsymbol{x}_{\mathrm{mis}} \mid \boldsymbol{x}_{\mathrm{obs}})}[p_{\boldsymbol{\theta}}(\boldsymbol{z} \mid \boldsymbol{x})]$, we propose using variational-mixtures to improve the match between the variational and model posteriors when dealing with incomplete data in order to improve model estimation performance.

**Marginalised variational bound.** In the standard ELBO derivation for incomplete data in eq. (2) the missing variables are first marginalised (collapsed) from the likelihood, and then a variational ELBO is established. This approach is sometimes referred to as collapsed variational inference (CVI). In contrast, in the derivation of the DeMissVAE encoder objective in eq. (12) we swap the order of marginalisation and variational inference. Specifically, we start with the variational ELBO on completed-data, and then marginalise the missing variables (see appendix B). This approach bears similarity to the marginalised variational bound (MVB, or KL-corrected bound) in exponential-conjugate variational inference literature (King & Lawrence, 2006; Lázaro-Gredilla & Titsias, 2011; Hensman et al., 2012). In these works, MVB has been preferred over CVI due to improved convergence and guarantees that, for appropriately formulated conjugate models, MVB is analytically tractable in cases where CVI is not (Hensman et al., 2012, Section 3.3). While MVB remains intractable in the VAE setting with incomplete data, similar to how the standard ELBO is intractable in fully-observed case, we find the motivation behind MVB and DeMissVAE to be similar.

## 6 Evaluation

We here evaluate the proposed methods, MissVAE, MissSVAE, MissIWAE, MissSIWAE (section 4.1), and DeMissVAE (section 4.2), on synthetic and real-world data, and compare them to the popular methods MVAE and MIWAE that do not use mixture variational distributions (Mattei & Frellsen, 2019). The methods are summarised in table 1 and the code implementation is available at https://github.com/vsimkus/demiss-vae.

### 6.1 Mixture-of-Gaussians data with a 2D latent VAE

Evaluating log-likelihood on held-out data is generally intractable for VAEs due to the intractable integral in eq. (1). We hence here choose a VAE with 2D latent space, where numerical integration can be used to estimate the log-likelihood of the model accurately (see appendix E.1 for more details). We fit the model on incomplete data drawn from a mixture-of-Gaussians distribution. By introducing uniform missingness of 50% in the mixture-of-Gaussians data we introduce multi-modality in the latent space (see fig. 1), which allows us to verify the efficacy of mixture-variational distributions when the posteriors are multi-modal due to missing data.

Results are shown in fig. 2. We first note that the stratified MissSVAE approach performed better than MissVAE that uses ancestral sampling. The reason for this is likely that stratified sampling reduces Monte Carlo variance of the gradients w.r.t. $\boldsymbol{\phi}$ and hence enables a better fit of the variational distribution $q_{\boldsymbol{\phi}}(\boldsymbol{z} \mid$

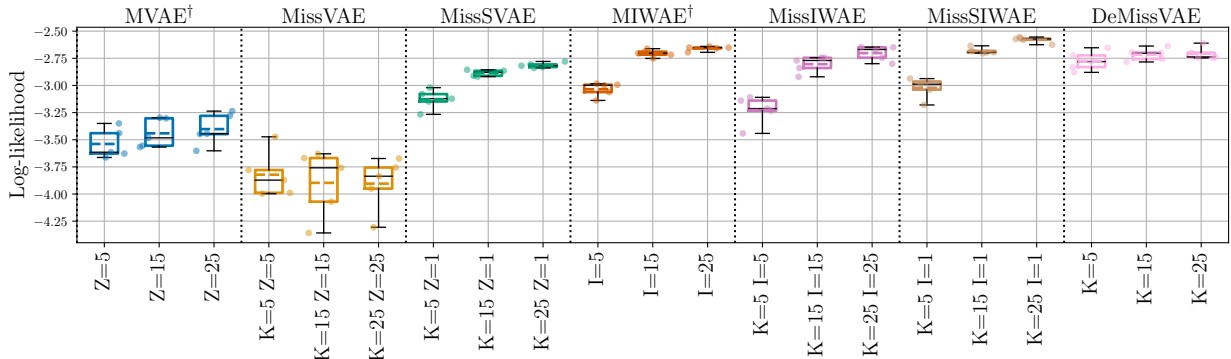

Figure 2: *Log-likelihood on held out data evaluated by numerically integrating the 2D latent variables.* VAEs were fitted on mixture-of-Gaussians data with 50% missingness. Each model is fitted with a computational budget of 5/15/25 samples from the variational distribution. The box plots show 1st and 3rd quartiles, the black lines are the medians, the dashed lines are the means, and the whiskers show the data range over 5 independent runs. MVAE and MIWAE (†) are baseline methods by Mattei & Frellsen (2019). The other five methods are proposed in this paper.

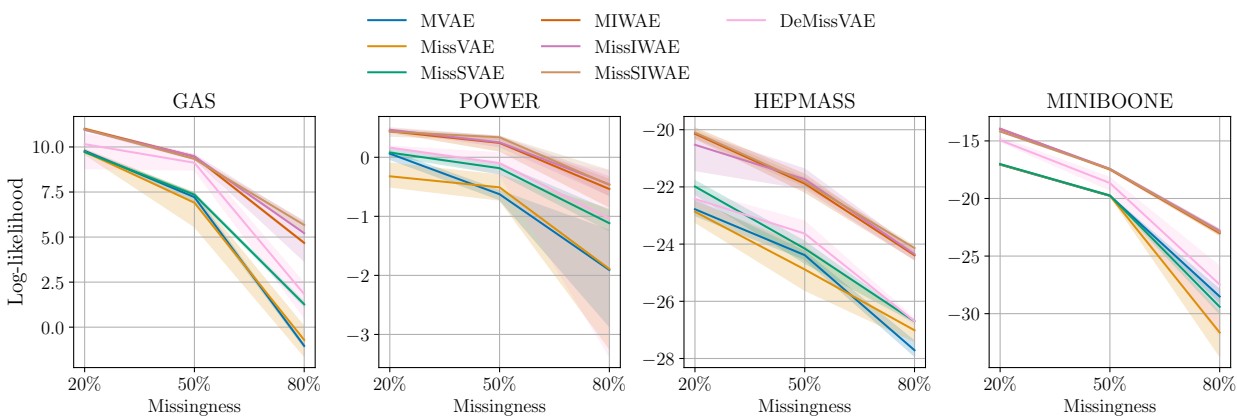

Figure 3: *Estimate of the test log-likelihood using the IWELBO with $I = 50000$, on four UCI data sets.* Each data set was rendered incomplete by applying uniform missingness of 20/50/80%. The curves show average performance over 5 independent runs of the algorithms and the intervals show the 90% centered interval.

$x_{\text{obs}}$) (see a further investigation in appendix F.1.1). In line with this intuition, the MissVAE results exhibit significantly larger variance than MissSVAE. Similarly, we observe that the stratified MissSIWAE approach performed better than MissIWAE. Importantly, we see that the use of mixture variational distributions in MissSVAE and MissSIWAE improve the model fit over the MVAE and MIWAE baselines that do not use mixtures to deal with the increased posterior complexity due to missingness. Finally, we observe that DeMissVAE is capable of achieving comparable performance to MIWAE and MissSIWAE, despite using a looser ELBO bound, which shows that the decomposed approach to handling data missingness can be used to achieve an improved fit of the model.

In appendix F.1.2, we analyse the model and variational posteriors of the learnt models. We observe that the mixture approaches better-approximate the incomplete-data posteriors, compared to the approaches that do not use variational-mixtures. Moreover, we also observe that the structure of the latent space is better-behaved when fitted using the decomposed approach in DeMissVAE.

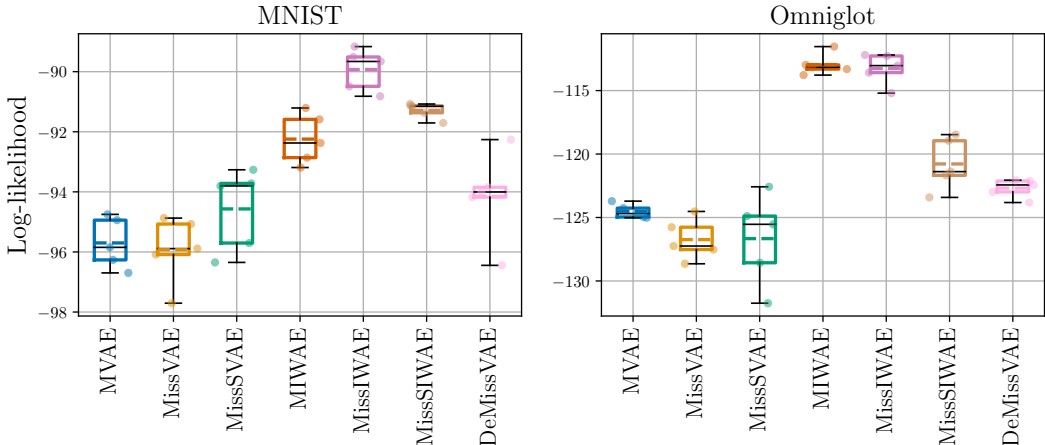

Figure 4: *Estimate of the test log-likelihood using the IWELBO with $I = 1000$, MNIST and Omniglot data sets.* Each image in the training data set was missing 2 out of 4 random quadrants. The box plots show 1st and 3rd quartiles, the black lines are the medians, the dashed lines are the means, and the whiskers show the data range over 5 independent runs.

## 6.2 Real-world UCI data sets

We here evaluate the proposed methods on real-world data sets from the UCI repository (Dua & Graff, 2017; Papamakarios et al., 2017). We train a VAE model with ResNet architecture on incomplete data sets with 20/50/80% uniform missingness (see appendix E.2 for more details). We then estimate the log-likelihood on complete test data set using the IWELBO bound with $I = 50K$ importance samples.[8] For additional metrics see appendix F.2.

The results are shown in fig. 3. We first note that, similar to before, the stratified MissSVAE approach performed better than MissVAE which uses ancestral sampling. Importantly, we observe that using mixture variational distributions in MissSVAE improves the fit of the model over MVAE (with the exception on the Miniboone data set) that uses non-mixture variational distributions. Furthermore, the gains in model accuracy typically increase with data missingness, which verifies that MissSVAE performs better because it handles the increased posterior complexity due to missing data better (see fig. 1). Next, we observe that the performance of MIWAE, MissIWAE, and MissSIWAE is similar, although we can note a small improvement by using MissIWAE and MissSIWAE in large missingness settings. We observe only a relatively small difference between the IWAE methods because the use of importance weighted bound already corresponds to using a more flexible semi-implicitly defined variational distribution (Cremer et al., 2017), which here seems to be sufficient to deal with the complexities arising due to missingness. Finally, we note that DeMissVAE results are in-between MissSVAE and MIWAE. This verifies that the decomposed approach can be used to deal with data missingness and, as a result, can improve the fit of the model. Nonetheless, DeMissVAE is surpassed by the IWAE methods, which is likely due to using the ELBO in DeMissVAE versus IWELBO in IWAE methods that can tighten the bound more effectively.

## 6.3 MNIST and Omniglot data sets

In this section we evaluate the proposed methods on binarised MNIST (Garris et al., 1991) and Omniglot (Lake et al., 2015) data sets of handwritten characters. We fit a VAE model with a convolutional ResNet encoder and decoder networks (see appendix E.3 for more details). The data is made incomplete by masking 2 out of 4 quadrants of an image at random. Similar to the previous section, we estimate the log-likelihood

---

[8]As $I \to \infty$ IWELBO approaches $\log p_{\boldsymbol{\theta}}(\boldsymbol{x})$. Moreover, as suggested by Mattei & Frellsen (2018b), to improve the estimate on held-out data we fine-tune the encoder on complete test data before estimating the log-likelihood.

on a complete test data set using the IWELBO bound with $I = 1000$ importance samples. In appendix F.3 we report additional results with varying dimensionality of the latent variables.

On the MNIST data set we see that MVAE $\leq$ MissVAE $<$ MissSVAE similar to the previous results but MIWAE $<$ MissSIWAE $<$ MissIWAE. This suggests that MissIWAE, which uses ancestral sampling, was able to tighten the bound more effectively compared to stratified MissSIWAE, and was able to fit the variational distribution $q_\phi(z \mid x_{\text{obs}})$ well despite the potentially larger variance w.r.t. $\phi$. Moreover, we also see that MVAE $<$ DeMissVAE $<$ MIWAE, which further verifies that the decomposed approach is able to handle the data missingness well.

On the Omniglot data we observe that the mixture approaches perform similarly to MVAE and MIWAE, which do not use mixture variational distributions. This suggests that either the posterior multi-modality is less prominent in the Omniglot data set or that due to the reverse KL optimisation of the variational distribution all mixture components have degenerated to a single mode. Finally, DeMissVAE slightly outperforms MVAE, MissVAE, and MissSVAE, but is surpassed by the importance-weighted approaches.

Interestingly, in this evaluation the stratified approaches (MissSVAE and MissSIWAE) were outperformed by the approaches using standard ELBO and implicit reparametrisation (MissVAE and MissIWAE). This suggests that the performance of each approach can be data- and model-dependent and hence both should be evaluated when possible.

## 7  Discussion

Handling missing data is a key challenge in modern machine learning, as many real-world applications involve incomplete data. In the context of variational autoencoders, we have shown that incomplete data increases the complexity of the latent variables' posterior distribution. Therefore, accurately fitting models from incomplete data requires more flexible variational families than in the complete-data case. We stipulated that variational-mixtures are a natural approach for handling missing data. One benefit is that it allows us to work with the same variational families as in the fully-observed case, which enables the transfer of useful known inductive biases (Miao et al., 2022) from the fully-observed to the incomplete data scenario.

Subsequently, we have introduced two methodologies grounded in variational mixtures. First, we proposed using finite variational mixtures with the standard and importance-weighted ELBOs using ancestral and stratified sampling of the mixtures. Second, we have proposed a novel "decomposed" variational-mixture approach, that uses cost-effective yet often coarse conditional sampling methods for VAEs to generate imputations and ELBO-based objectives that are robust to the sampling errors.

Our evaluation shows that using variational mixtures can improve the fit of VAEs when dealing with incomplete data, surpassing the performance of models without variational mixtures. Moreover, our observations indicate that, although stratified sampling of the finite mixtures often yields better results compared to ancestral sampling, the effectiveness of these methods can be data- and model-dependent and hence both approaches should be evaluated when possible. Our results further indicate that variational mixtures provide relatively little improvement with the IWELBO-based methods compared the ELBO-based methods. We believe that this is mainly because IWELBO can be seen to be working with semi-implicitly defined variational distributions that are flexible enough to handle the posterior complexity increase due to missing data. Alternatively, this may be related to an observation by Shi et al. (2019, Appendix A) that the reverse-KL formulation of the importance-weighted bound may lead to situations where the mixture components collapse to a single mode. Hence, a future direction would be to investigate alternative formulations of importance-weighted bounds that avoid the mode-seeking nature (Bornschein & Bengio, 2015; Mnih & Rezende, 2016; Wan et al., 2020). Furthermore, we note that the decomposed approach in DeMissVAE outperforms all ELBO-based methods but falls short of surpassing IWELBO-based methods. These results point towards promising research avenues, suggesting potential improvements in VAE model estimation from incomplete data. Future directions include extending the DeMissVAE approach to incorporate IWELBO-based objectives and developing improved cost-effective conditional sampling methods for VAEs.

| Method | Budget | Variational families | Latent structure* | Evaluation rank | | |
|---|---|---|---|---|---|---|
| | | | | *MoG* | *UCI* | *MNIST+Omniglot* |
| MissVAE | *small* | *limited* | *well-behaved* | 5 | 5 | 5 |
| MissSVAE | *medium* | *any* | *well-behaved* | 4 | 4 | 4 |
| MissIWAE | *medium* | *limited* | *potentially irregular* | 3 | 2 | 1 |
| MissSIWAE | *medium* | *any* | *potentially irregular* | 1 | 1 | 2 |
| DeMissVAE | *medium/high* | *any* | *well-behaved$^+$* | 2 | 3 | 3 |

Table 2: *A coarse summary of advantages and disadvantages of the proposed methods. Budget*: small/medium/high depending on the number of latent samples required or whether conditional sampling of VAEs is needed. *Variational families*: which families of distributions can be used as mixture components— any reparametrisable families, or limited families, as discussed in section 4.1. *Latent structure*: methods with potential to learn irregular latent spaces may have decreased downstream performance in certain tasks. We have found (+) that DeMissVAE is able to achieve the most well-behaved latent structures on the MoG data in appendix F.1.2. Please note (*) that the learnt latent structure will depend on the chosen model architecture. *Evaluation rank*: the rank of the proposed methods in the evaluations in sections 6.1 to 6.3.

The choice between the proposed methods for fitting VAEs from incomplete data depends on various factors such as computational budget, variational families, model accuracy goals, and the specific requirements of downstream tasks, discussed next and summarised in table 2.

**Computational and memory budget.** The standard ELBO with ancestral sampling is the most suitable method for small computational and memory budgets, since the objective can be estimated using a single latent sample for each data point. On the other hand, methods using stratified sampling or the importance-weighted ELBO require multiple latent samples for each data-point and hence may only be used if the memory and compute budget allows. Moreover, for a fixed budget, stratified approaches may limit the number of components $K$ that may be used. Lastly, akin to the standard ELBO, the DeMissVAE objectives can be estimated using a single latent sample, but the approach incurs extra cost in sampling the imputations.

**Variational families.** While the stratified and DeMissVAE approaches can use any reparametrisable distribution family for the mixture components, the ancestral sampling methods require the use of *implicit* reparametrisation (Figurnov et al., 2019) and as a result may not work with all distribution families (see discussion in section 4.1).

**Model accuracy.** Stratified sampling of mixtures can improve the model accuracy, compared to ancestral sampling, by reducing Monte Carlo gradient variance. Additionally, methods using the importance-weighted ELBO, compared to the standard ELBO, are often able to tighten the bound more effectively by using multiple importance samples, leading to improved model accuracy. DeMissVAE performance lies in between the standard ELBO and importance-weighted ELBO approaches. Although the introduced DeMissVAE objectives exhibit robustness to some imputation distribution error, improved model accuracy can often be achieved by improving the accuracy of imputations by using a larger budget for the imputation step.

**Latent structure.** Different downstream tasks may prefer distinct latent structures, for example, conditional generation from unconditional VAEs is often easier if the latent space is well-structured (Engel et al., 2017; Gómez-Bombarelli et al., 2018). To this end, observations in appendix F.1.2 show that the latent space of DeMissVAE behaves well, and is comparable to a model fitted with complete data. This characteristic makes it preferable for downstream tasks requiring well-structured latent spaces. On the other hand, as noted by Burda et al. (2015, Appendix C) and Cremer et al. (2018, Section 5.4), the use of importance-weighted ELBO to mitigate the increased posterior complexity due to missing data may make the latent space less regular, compared to a model trained on fully-observed data set, which potentially decreases the model's performance on downstream tasks.

Finally, we step back to note that this paper is focused on the class of variational autoencoder models, a subset of the broader family of deep latent variable models (DLVMs). Much like VAEs, DLVMs usually aim to efficiently represent the intricate nature of data through a well-structured latent space, implicitly defined by a learnable generative process. Building on our findings in VAEs, where incomplete data led to an increased complexity in the posterior distribution compared to the fully-observed case, we conjecture that a similar effect may occur within the wider family of DLVMs, affecting the fit of the model. We therefore believe that there is substantial scope to explore the implications of incomplete data in other DLVM classes, particularly focusing on the effects of marginalisation on latent space representations and the associated generative processes. Investigating decomposed approaches, similar to DeMissVAE or Monte Carlo EM (Wei & Tanner, 1990), presents promising avenues for further research in this direction.

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

# A   Posterior complexity due to missing information

The complexity increase of the model posterior due to missing data, shown in fig. 1, explains why flexible variational distributions (Burda et al., 2015; Cremer et al., 2017) have been preferred when fitting VAEs from incomplete data (Mattei & Frellsen, 2019; Ipsen et al., 2020; Ma & Zhang, 2021). We here define the increase of the posterior complexity via the expected Kullback–Leibler (KL) divergence as follows

$$\mathbb{E}_{p^*(\boldsymbol{x})}\left[D_{\mathrm{KL}}(p_{\boldsymbol{\theta}}(\boldsymbol{z}\mid\boldsymbol{x})\,\|\,p_{\boldsymbol{\theta}}(\boldsymbol{z}\mid\boldsymbol{x}_{\mathrm{obs}}))\right]=\mathbb{E}_{p^*(\boldsymbol{x})}\mathbb{E}_{p_{\boldsymbol{\theta}}(\boldsymbol{z}|\boldsymbol{x})}\left[\log\frac{p_{\boldsymbol{\theta}}(\boldsymbol{z}\mid\boldsymbol{x}_{\mathrm{obs}},\boldsymbol{x}_{\mathrm{mis}})}{p_{\boldsymbol{\theta}}(\boldsymbol{z}\mid\boldsymbol{x}_{\mathrm{obs}})}\right]=\mathcal{I}(\boldsymbol{z},\boldsymbol{x}_{\mathrm{mis}}\mid\boldsymbol{x}_{\mathrm{obs}}).[9]$$

As shown above the expected KL divergence equals the (conditional) mutual information (MI) between the latents $\boldsymbol{z}$ and the missing variables $\boldsymbol{x}_{\mathrm{mis}}$.

The mutual information interpretation allows us to reason when a more flexible variational family may be necessary to accurately estimate VAEs from incomplete data. Specifically, when the MI is small then the two posterior distributions, $p_{\boldsymbol{\theta}}(\boldsymbol{z}\mid\boldsymbol{x})$ and $p_{\boldsymbol{\theta}}(\boldsymbol{z}\mid\boldsymbol{x}_{\mathrm{obs}})$ are similar, in which case a simple variational distribution may work sufficiently well. This situation might appear when the observed $\boldsymbol{x}_{\mathrm{obs}}$ and unobserved $\boldsymbol{x}_{\mathrm{mis}}$ variables are highly related and $\boldsymbol{x}_{\mathrm{mis}}$ provides little additional information about $\boldsymbol{z}$ over just $\boldsymbol{x}_{\mathrm{obs}}$, for example, when random pixels of an image are masked it is "easy" to infer the complete image due to strong relationship between neighbouring pixels. On the other hand, when the MI is high then $\boldsymbol{x}_{\mathrm{mis}}$ provides significant additional information about $\boldsymbol{z}$ over just $\boldsymbol{x}_{\mathrm{obs}}$, in which case a more flexible variational family may be needed, for example, when the pixels of an image are masked in blocks such that it introduces significant uncertainty about what is missing.

# B   DeMissVAE: Encoder objective derivation

The standard (complete-data) ELBO in eq. (2) gives the inequality

$$\log p_{\boldsymbol{\theta}}(\boldsymbol{x}_{\mathrm{obs}},\boldsymbol{x}_{\mathrm{mis}})\geq\mathbb{E}_{q_{\boldsymbol{\phi}}(\boldsymbol{z}|\boldsymbol{x}_{\mathrm{obs}},\boldsymbol{x}_{\mathrm{mis}})}\left[\log\frac{p_{\boldsymbol{\theta}}(\boldsymbol{x}_{\mathrm{obs}},\boldsymbol{x}_{\mathrm{mis}},\boldsymbol{z})}{q_{\boldsymbol{\phi}}(\boldsymbol{z}\mid\boldsymbol{x}_{\mathrm{obs}},\boldsymbol{x}_{\mathrm{mis}})}\right],$$

which, together with the identity

$$\log p_{\boldsymbol{\theta}}(\boldsymbol{x}_{\mathrm{obs}})=\log\int\exp\left\{\log p_{\boldsymbol{\theta}}(\boldsymbol{x}_{\mathrm{obs}},\boldsymbol{x}_{\mathrm{mis}})\right\}\mathrm{d}\boldsymbol{x}_{\mathrm{mis}},$$

yields

$$\log p_{\boldsymbol{\theta}}(\boldsymbol{x}_{\mathrm{obs}})\geq\log\int\exp\left\{\mathbb{E}_{q_{\boldsymbol{\phi}}(\boldsymbol{z}|\boldsymbol{x}_{\mathrm{obs}},\boldsymbol{x}_{\mathrm{mis}})}\left[\log\frac{p_{\boldsymbol{\theta}}(\boldsymbol{x}_{\mathrm{obs}},\boldsymbol{x}_{\mathrm{mis}},\boldsymbol{z})}{q_{\boldsymbol{\phi}}(\boldsymbol{z}\mid\boldsymbol{x}_{\mathrm{obs}},\boldsymbol{x}_{\mathrm{mis}})}\right]\right\}\mathrm{d}\boldsymbol{x}_{\mathrm{mis}}.$$

As the integral on the r.h.s. is intractable, we lower-bound it using the imputation distribution $f^t(\boldsymbol{x}_{\mathrm{mis}}\mid\boldsymbol{x}_{\mathrm{obs}})$ and Jensen's inequality

$$\log p_{\boldsymbol{\theta}}(\boldsymbol{x}_{\mathrm{obs}})\geq\log\int\frac{f^t(\boldsymbol{x}_{\mathrm{mis}}\mid\boldsymbol{x}_{\mathrm{obs}})}{f^t(\boldsymbol{x}_{\mathrm{mis}}\mid\boldsymbol{x}_{\mathrm{obs}})}\exp\left\{\mathbb{E}_{q_{\boldsymbol{\phi}}(\boldsymbol{z}|\boldsymbol{x}_{\mathrm{obs}},\boldsymbol{x}_{\mathrm{mis}})}\left[\log\frac{p_{\boldsymbol{\theta}}(\boldsymbol{x}_{\mathrm{obs}},\boldsymbol{x}_{\mathrm{mis}},\boldsymbol{z})}{q_{\boldsymbol{\phi}}(\boldsymbol{z}\mid\boldsymbol{x}_{\mathrm{obs}},\boldsymbol{x}_{\mathrm{mis}})}\right]\right\}\mathrm{d}\boldsymbol{x}_{\mathrm{mis}}$$

$$=\log\mathbb{E}_{f^t(\boldsymbol{x}_{\mathrm{mis}}|\boldsymbol{x}_{\mathrm{obs}})}\left[\exp\left(-\log f^t(\boldsymbol{x}_{\mathrm{mis}}\mid\boldsymbol{x}_{\mathrm{obs}})\right)\exp\left\{\mathbb{E}_{q_{\boldsymbol{\phi}}(\boldsymbol{z}|\boldsymbol{x}_{\mathrm{obs}},\boldsymbol{x}_{\mathrm{mis}})}\left[\log\frac{p_{\boldsymbol{\theta}}(\boldsymbol{x}_{\mathrm{obs}},\boldsymbol{x}_{\mathrm{mis}},\boldsymbol{z})}{q_{\boldsymbol{\phi}}(\boldsymbol{z}\mid\boldsymbol{x}_{\mathrm{obs}},\boldsymbol{x}_{\mathrm{mis}})}\right]\right\}\right]$$

$$=\log\mathbb{E}_{f^t(\boldsymbol{x}_{\mathrm{mis}}|\boldsymbol{x}_{\mathrm{obs}})}\left[\exp\left\{\mathbb{E}_{q_{\boldsymbol{\phi}}(\boldsymbol{z}|\boldsymbol{x}_{\mathrm{obs}},\boldsymbol{x}_{\mathrm{mis}})}\left[\log\frac{p_{\boldsymbol{\theta}}(\boldsymbol{x}_{\mathrm{obs}},\boldsymbol{x}_{\mathrm{mis}},\boldsymbol{z})}{q_{\boldsymbol{\phi}}(\boldsymbol{z}\mid\boldsymbol{x}_{\mathrm{obs}},\boldsymbol{x}_{\mathrm{mis}})f^t(\boldsymbol{x}_{\mathrm{mis}}\mid\boldsymbol{x}_{\mathrm{obs}})}\right]\right\}\right]$$

---

[9]Where notation of $\boldsymbol{m}$ is suppressed due to MAR assumption. In case of missing-not-at-random (MNAR) assumption there would be an additional dependency on $\boldsymbol{m}$.

$$\geq \mathbb{E}_{f^t(\boldsymbol{x}_{\mathrm{mis}}|\boldsymbol{x}_{\mathrm{obs}})q_{\boldsymbol{\phi}}(\boldsymbol{z}|\boldsymbol{x}_{\mathrm{obs}},\boldsymbol{x}_{\mathrm{mis}})}\left[\log\frac{p_{\boldsymbol{\theta}}(\boldsymbol{x}_{\mathrm{obs}},\boldsymbol{x}_{\mathrm{mis}},\boldsymbol{z})}{q_{\boldsymbol{\phi}}(\boldsymbol{z}\mid\boldsymbol{x}_{\mathrm{obs}},\boldsymbol{x}_{\mathrm{mis}})f^t(\boldsymbol{x}_{\mathrm{mis}}\mid\boldsymbol{x}_{\mathrm{obs}})}\right]$$

$$=\underbrace{\mathbb{E}_{f^t(\boldsymbol{x}_{\mathrm{mis}}|\boldsymbol{x}_{\mathrm{obs}})q_{\boldsymbol{\phi}}(\boldsymbol{z}|\boldsymbol{x}_{\mathrm{obs}},\boldsymbol{x}_{\mathrm{mis}})}\left[\log\frac{p_{\boldsymbol{\theta}}(\boldsymbol{x}_{\mathrm{obs}},\boldsymbol{x}_{\mathrm{mis}},\boldsymbol{z})}{q_{\boldsymbol{\phi}}(\boldsymbol{z}\mid\boldsymbol{x}_{\mathrm{obs}},\boldsymbol{x}_{\mathrm{mis}})}\right]}_{\overset{\mathrm{def}}{=}\mathcal{L}_{\mathrm{LMVB}}^{\boldsymbol{\phi}}(\boldsymbol{x}_{\mathrm{obs}};\boldsymbol{\phi},\boldsymbol{\theta},f^t)}+\underbrace{\mathcal{H}\left[f^t(\boldsymbol{x}_{\mathrm{mis}}\mid\boldsymbol{x}_{\mathrm{obs}})\right]}_{\text{Const. w.r.t. }\boldsymbol{\phi}}.$$

## C  DeMissVAE: Motivating the separation of objectives

The two DeMissVAE objectives $\mathcal{L}_{\mathrm{CVI}}^{\boldsymbol{\theta}}$ and $\mathcal{L}_{\mathrm{LMVB}}^{\boldsymbol{\phi}}$ in eqs. (10) and (12) correspond to valid lower-bounds on $\log p_{\boldsymbol{\theta}}(\boldsymbol{x}_{\mathrm{obs}})$ irrespective of $f^t(\boldsymbol{x}_{\mathrm{mis}}\mid\boldsymbol{x}_{\mathrm{obs}})$. Moreover, both of them are tight at $f^t(\boldsymbol{x}_{\mathrm{mis}}\mid\boldsymbol{x}_{\mathrm{obs}}) = p_{\boldsymbol{\theta}}(\boldsymbol{x}_{\mathrm{mis}}\mid\boldsymbol{x}_{\mathrm{obs}})$ and $q_{\boldsymbol{\phi}}(\boldsymbol{z}\mid\boldsymbol{x}_{\mathrm{obs}},\boldsymbol{x}_{\mathrm{mis}}) = p_{\boldsymbol{\theta}}(\boldsymbol{z}\mid\boldsymbol{x}_{\mathrm{obs}},\boldsymbol{x}_{\mathrm{mis}})$. *So, a natural question is why do we prefer $\mathcal{L}_{\mathrm{CVI}}^{\boldsymbol{\theta}}$ to learn $p_{\boldsymbol{\theta}}$ and $\mathcal{L}_{\mathrm{LMVB}}^{\boldsymbol{\phi}}$ to learn $q_{\boldsymbol{\phi}}$?*

**Why use $\mathcal{L}_{\mathrm{CVI}}^{\boldsymbol{\theta}}$ in eq. (10) over $\mathcal{L}_{\mathrm{LMVB}}^{\boldsymbol{\phi}}$ in eq. (12) to learn $p_{\boldsymbol{\theta}}(\boldsymbol{x})$?**  Maximisation of the objective $\mathcal{L}_{\mathrm{LMVB}}^{\boldsymbol{\phi}}$ in iteration $t$ w.r.t. $\boldsymbol{\theta}$ would have to compromise between maximising the log-likelihood $\log p_{\boldsymbol{\theta}}(\boldsymbol{x}_{\mathrm{obs}})$ and keeping the other two KL divergence terms in eq. (13) low. Specifically, the compromise between maximising $\log p_{\boldsymbol{\theta}}(\boldsymbol{x}_{\mathrm{obs}})$ and keeping $D_{\mathrm{KL}}(f^t(\boldsymbol{x}_{\mathrm{mis}}\mid\boldsymbol{x}_{\mathrm{obs}})\mid\mid p_{\boldsymbol{\theta}}(\boldsymbol{x}_{\mathrm{mis}}\mid\boldsymbol{x}_{\mathrm{obs}}))$ low is equivalent to the compromise in the EM algorithm, which is known to affect the convergence of the model (Meng, 1994). Moreover, if $f^t(\boldsymbol{x}_{\mathrm{mis}}\mid\boldsymbol{x}_{\mathrm{obs}}) \neq p_{\boldsymbol{\theta}}(\boldsymbol{x}_{\mathrm{mis}}\mid\boldsymbol{x}_{\mathrm{obs}})$ then minimising the $D_{\mathrm{KL}}(f^t(\boldsymbol{x}_{\mathrm{mis}}\mid\boldsymbol{x}_{\mathrm{obs}})\mid\mid p_{\boldsymbol{\theta}}(\boldsymbol{x}_{\mathrm{mis}}\mid\boldsymbol{x}_{\mathrm{obs}}))$ will fit the model $p_{\boldsymbol{\theta}}(\boldsymbol{x})$ to the biased samples from $f^t(\boldsymbol{x}_{\mathrm{mis}}\mid\boldsymbol{x}_{\mathrm{obs}})$. On the other hand, in $\mathcal{L}_{\mathrm{CVI}}^{\boldsymbol{\theta}}$ the missing variables $\boldsymbol{x}_{\mathrm{mis}}$ are marginalised from the model, therefore it avoids the compromise with $D_{\mathrm{KL}}(f^t(\boldsymbol{x}_{\mathrm{mis}}\mid\boldsymbol{x}_{\mathrm{obs}})\mid\mid p_{\boldsymbol{\theta}}(\boldsymbol{x}_{\mathrm{mis}}\mid\boldsymbol{x}_{\mathrm{obs}}))$ and the potential bias of the imputation distribution $f^t(\boldsymbol{x}_{\mathrm{mis}}\mid\boldsymbol{x}_{\mathrm{obs}})$ affects the model *only* via the latents $\boldsymbol{z}\sim q_{\boldsymbol{\phi},f^t}(\boldsymbol{z}\mid\boldsymbol{x}_{\mathrm{obs}})$, increasing the robustness to sub-optimal imputations.

**Why use $\mathcal{L}_{\mathrm{LMVB}}^{\boldsymbol{\phi}}$ in eq. (12) over $\mathcal{L}_{\mathrm{CVI}}^{\boldsymbol{\theta}}$ in eq. (10) to learn $q_{\boldsymbol{\phi}}(\boldsymbol{z}\mid\boldsymbol{x})$?**  In the case of $\mathcal{L}_{\mathrm{CVI}}^{\boldsymbol{\theta}}$, if $f^t(\boldsymbol{x}_{\mathrm{mis}}\mid\boldsymbol{x}_{\mathrm{obs}}) = p_{\boldsymbol{\theta}}(\boldsymbol{x}_{\mathrm{mis}}\mid\boldsymbol{x}_{\mathrm{obs}})$ then the bound is tightened when $q_{\boldsymbol{\phi}}(\boldsymbol{z}\mid\boldsymbol{x}_{\mathrm{obs}},\boldsymbol{x}_{\mathrm{mis}}) = p_{\boldsymbol{\theta}}(\boldsymbol{z}\mid\boldsymbol{x}_{\mathrm{obs}},\boldsymbol{x}_{\mathrm{mis}})$ for all $\boldsymbol{x}_{\mathrm{mis}}$, which is the same optimal $q_{\boldsymbol{\phi}}$ if we used $\mathcal{L}_{\mathrm{LMVB}}^{\boldsymbol{\phi}}$. But, there is also at least one more possible optimal solution $q_{\boldsymbol{\phi}}(\boldsymbol{z}\mid\boldsymbol{x}_{\mathrm{obs}},\boldsymbol{x}_{\mathrm{mis}}) = p_{\boldsymbol{\theta}}(\boldsymbol{z}\mid\boldsymbol{x}_{\mathrm{obs}})$, which ignores the imputations and corresponds to the optimal solution of the standard approach in section 2, and thus it means that the optimum is (partially) unidentifiable and can make optimisation of $q_{\boldsymbol{\phi}}$ using $\mathcal{L}_{\mathrm{CVI}}^{\boldsymbol{\theta}}$ difficult. Moreover, if $f^t(\boldsymbol{x}_{\mathrm{mis}}\mid\boldsymbol{x}_{\mathrm{obs}}) \neq p_{\boldsymbol{\theta}}(\boldsymbol{x}_{\mathrm{mis}}\mid\boldsymbol{x}_{\mathrm{obs}})$ then in order to minimise $D_{\mathrm{KL}}(q_{\boldsymbol{\phi},f^t}(\boldsymbol{z}\mid\boldsymbol{x}_{\mathrm{obs}})\mid\mid p_{\boldsymbol{\theta}}(\boldsymbol{z}\mid\boldsymbol{x}_{\mathrm{obs}}))$ w.r.t. $\boldsymbol{\phi}$ the variational distribution $q_{\boldsymbol{\phi}}(\boldsymbol{z}\mid\boldsymbol{x}_{\mathrm{obs}},\boldsymbol{x}_{\mathrm{mis}})$ would have to compensate for the inaccuracies of $f^t(\boldsymbol{x}_{\mathrm{mis}}\mid\boldsymbol{x}_{\mathrm{obs}})$ by adjusting the probability mass over the latents $\boldsymbol{z}$, such that $q_{\boldsymbol{\phi}}(\boldsymbol{z}\mid\boldsymbol{x}_{\mathrm{obs}},\boldsymbol{x}_{\mathrm{mis}})$ is correct on average, i.e. $q_{\boldsymbol{\phi},f^t}(\boldsymbol{z}\mid\boldsymbol{x}_{\mathrm{obs}}) = \mathbb{E}_{f^t(\boldsymbol{x}_{\mathrm{mis}}|\boldsymbol{x}_{\mathrm{obs}})}[q_{\boldsymbol{\phi}}(\boldsymbol{z}\mid\boldsymbol{x}_{\mathrm{obs}},\boldsymbol{x}_{\mathrm{mis}})] \approx p_{\boldsymbol{\theta}}(\boldsymbol{z}\mid\boldsymbol{x}_{\mathrm{obs}})$. These two issues make optimising $\boldsymbol{\phi}$ via $\mathcal{L}_{\mathrm{CVI}}^{\boldsymbol{\theta}}$ such that $q_{\boldsymbol{\phi}}(\boldsymbol{z}\mid\boldsymbol{x}_{\mathrm{obs}},\boldsymbol{x}_{\mathrm{mis}}) \approx p_{\boldsymbol{\theta}}(\boldsymbol{z}\mid\boldsymbol{x}_{\mathrm{obs}},\boldsymbol{x}_{\mathrm{mis}})$ difficult. On the other hand, in $\mathcal{L}_{\mathrm{LMVB}}^{\boldsymbol{\phi}}$ the optimal $q_{\boldsymbol{\phi}}$ is always at $q_{\boldsymbol{\phi}}(\boldsymbol{z}\mid\boldsymbol{x}_{\mathrm{obs}},\boldsymbol{x}_{\mathrm{mis}}) = p_{\boldsymbol{\theta}}(\boldsymbol{z}\mid\boldsymbol{x}_{\mathrm{obs}},\boldsymbol{x}_{\mathrm{mis}})$, irrespective of the imputation distribution $f^t(\boldsymbol{x}_{\mathrm{mis}}\mid\boldsymbol{x}_{\mathrm{obs}})$, hence the $\mathcal{L}_{\mathrm{LMVB}}^{\boldsymbol{\phi}}$ objective in eq. (12) is well-defined and more robust to inaccuracies of $f^t(\boldsymbol{x}_{\mathrm{mis}}\mid\boldsymbol{x}_{\mathrm{obs}})$ for the optimisation of $q_{\boldsymbol{\phi}}(\boldsymbol{z}\mid\boldsymbol{x}_{\mathrm{obs}},\boldsymbol{x}_{\mathrm{mis}})$.

In fig. 5 we verify the efficacy of DeMissVAE via a control study on a small VAE model $p_{\boldsymbol{\theta}}(\boldsymbol{x})$ with 2D latent space fitted on incomplete samples from a ground truth mixture-of-Gaussians (MoG) distribution $p^*(\boldsymbol{x})$. We evaluate fitting the VAE using only $\mathcal{L}_{\mathrm{CVI}}^{\boldsymbol{\theta}}$ in eq. (10) (CVI-VAE, blue), only $\mathcal{L}_{\mathrm{LMVB}}^{\boldsymbol{\phi}}$ in eq. (12) (MVB-VAE, yellow), and using the proposed two-objective approach (DeMissVAE, green). In the left-most figure we evaluate the three methods where we represent the imputation distribution $f^t(\boldsymbol{x}_{\mathrm{mis}}\mid\boldsymbol{x}_{\mathrm{obs}}) = p_{\boldsymbol{\theta}}(\boldsymbol{x}_{\mathrm{mis}}\mid\boldsymbol{x}_{\mathrm{obs}})$ using rejection sampling, which corresponds to the optimal imputation distribution w.r.t. $D_{\mathrm{KL}}(f^t(\boldsymbol{x}_{\mathrm{mis}}\mid\boldsymbol{x}_{\mathrm{obs}})\mid\mid p_{\boldsymbol{\theta}}(\boldsymbol{x}_{\mathrm{mis}}\mid\boldsymbol{x}_{\mathrm{obs}})) = 0$. We see that the proposed approach (green) dominates over the other two control methods (blue and yellow), and importantly that marginalisation of the missing variables in DeMissVAE (green) improves the model accuracy compared to an EM-type handling of the missing variables (yellow). Furthermore, in the remaining two figures we investigate the sensitivity of the methods to the accuracy of imputations in $f^t(\boldsymbol{x}_{\mathrm{mis}}\mid\boldsymbol{x}_{\mathrm{obs}})$. In Oracle 1 we start with the ground-truth conditional

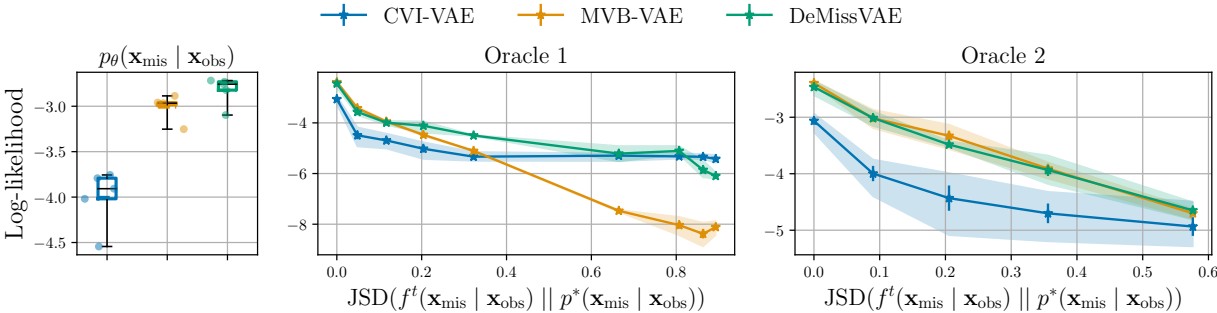

Figure 5: *A control study on a VAE model with 2D latent space (see additional details in appendix E.1), examining the sensitivity of the proposed method (DeMissVAE, green) and two control methods (blue and yellow) to the accuracy of the imputation distribution $f^t(\boldsymbol{x}_{\text{mis}} \mid \boldsymbol{x}_{\text{obs}})$. Left: $f^t(\boldsymbol{x}_{\text{mis}} \mid \boldsymbol{x}_{\text{obs}}) = p_{\boldsymbol{\theta}}(\boldsymbol{x}_{\text{mis}} \mid \boldsymbol{x}_{\text{obs}})$ represented using rejection sampling. Center: an oracle imputation function that gets progressively "wider" from left-to-right of the figure. Right: an oracle imputation distribution that towards the right of the figure more significantly oversamples low-probability posterior modes. The log-likelihood is computed on a held-out test data set by numerically integrating the 2D latent space of the VAE. The horizontal axis on the two right-most figures shows the Jensen–Shannon divergence between the imputation distribution and the ground-truth conditional $p^*(\boldsymbol{x}_{\text{mis}} \mid \boldsymbol{x}_{\text{obs}})$.*

$p^*(\boldsymbol{x}_{\text{mis}} \mid \boldsymbol{x}_{\text{obs}})$ and, along the x-axis of the figure, investigate how the methods perform when the imputation distribution becomes "wider": first interpolating from $p^*(\boldsymbol{x}_{\text{mis}} \mid \boldsymbol{x}_{\text{obs}})$ to an independent unconditional distribution $\prod_{d \in \text{idx}(\boldsymbol{m})} p^*(x_d)$ and then further towards and independent Gaussian distribution. And in Oracle 2 we investigate what happens when the sampler "oversamples" posterior modes: we interpolate the imputation distribution from $p^*(\boldsymbol{x}_{\text{mis}} \mid \boldsymbol{x}_{\text{obs}})$ to $\frac{1}{C} \sum_c^C p^*(\boldsymbol{x}_{\text{mis}} \mid \boldsymbol{x}_{\text{obs}}, c)$, where $c$ is the component of the mixture distribution with a total of $C$ components. As we see in the figure, the proposed DeMissVAE approach (green) performs similar or better than the MVB-VAE (yellow) and CVI-VAE (blue) control methods, with an exception when the $f^t(\boldsymbol{x}_{\text{mis}} \mid \boldsymbol{x}_{\text{obs}})$ are extremely inaccurate (last two points on the middle figure) which is expected since $q_{\boldsymbol{\phi}, f^t}(\boldsymbol{z} \mid \boldsymbol{x}_{\text{obs}})$ in eq. (9) can be arbitrarily far from $p_{\boldsymbol{\theta}}(\boldsymbol{z} \mid \boldsymbol{x}_{\text{obs}})$ when $q_{\boldsymbol{\phi}}(\boldsymbol{z} \mid \boldsymbol{x}_{\text{obs}}, \boldsymbol{x}_{\text{mis}}) = p_{\boldsymbol{\theta}}(\boldsymbol{z} \mid \boldsymbol{x}_{\text{obs}}, \boldsymbol{x}_{\text{mis}})$ but $f^t(\boldsymbol{x}_{\text{mis}} \mid \boldsymbol{x}_{\text{obs}}) \not\approx p_{\boldsymbol{\theta}}(\boldsymbol{x}_{\text{mis}} \mid \boldsymbol{x}_{\text{obs}})$.

Finally, in fig. 6 we investigate what happens if we used only $\mathcal{L}_{\text{CVI}}^{\boldsymbol{\theta}}$ in eq. (10) or $\mathcal{L}_{\text{LMVB}}^{\boldsymbol{\phi}}$ in eq. (12) to fit the VAE model, in contrast to the two separate objectives for encoder and decoder in DeMissVAE. We use the LAIR sampling method (Simkus & Gutmann, 2023) as detailed in appendix D to obtain approximate samples from $f^t(\boldsymbol{x}_{\text{mis}} \mid \boldsymbol{x}_{\text{obs}})(\boldsymbol{x}_{\text{mis}} \mid \boldsymbol{x}_{\text{obs}}) \approx p_{\boldsymbol{\theta}}(\boldsymbol{x}_{\text{mis}} \mid \boldsymbol{x}_{\text{obs}})$. And, we observe that DeMissVAE achieves a better fit of the model, in line with our motivation in this section.

## D  DeMissVAE: Implementing the training procedure

DeMissVAE requires optimising two objectives $\mathcal{L}_{\text{CVI}}^{\boldsymbol{\theta}}$ and $\mathcal{L}_{\text{LMVB}}^{\boldsymbol{\phi}}$ in eqs. (10) and (12) and drawing (approximate) samples to represent $f^t(\boldsymbol{x}_{\text{mis}} \mid \boldsymbol{x}_{\text{obs}}) \approx p_{\boldsymbol{\theta}}(\boldsymbol{x}_{\text{mis}} \mid \boldsymbol{x}_{\text{obs}})$. Our aim is to implement this efficiently to minimise redundant computation.

The algorithm starts with a randomly-initialised target VAE model $p_{\boldsymbol{\theta}}(\boldsymbol{x}, \boldsymbol{z})$, an amortised variational distribution $q_{\boldsymbol{\phi}}(\boldsymbol{z} \mid \boldsymbol{x})$, and an incomplete data set $\mathcal{D} = \{\boldsymbol{x}_{\text{obs}}^i\}_i$. And then, to represent the imputation distribution $f^0(\boldsymbol{x}_{\text{mis}} \mid \boldsymbol{x}_{\text{obs}})$, $K$ imputations $\{\boldsymbol{x}_{\text{mis}}^{ik}\}_{k=1}^K$ are generated for each $\boldsymbol{x}_{\text{obs}}^i \in \mathcal{D}$ using some simple imputation function such as sampling the marginal empirical distributions of the missing variables. The algorithm then iterates between the following two steps:

1. **Imputation.** Update the $K$ imputations $\{\boldsymbol{x}_{\text{mis}}^{ik}\}_{k=1}^K$ representing samples from the imputation distribution $f^t(\boldsymbol{x}_{\text{mis}} \mid \boldsymbol{x}_{\text{obs}})$, such that $f^t(\boldsymbol{x}_{\text{mis}} \mid \boldsymbol{x}_{\text{obs}})$ is "closer" to $p_{\boldsymbol{\theta}}(\boldsymbol{x}_{\text{mis}} \mid \boldsymbol{x}_{\text{obs}})$. For this, we

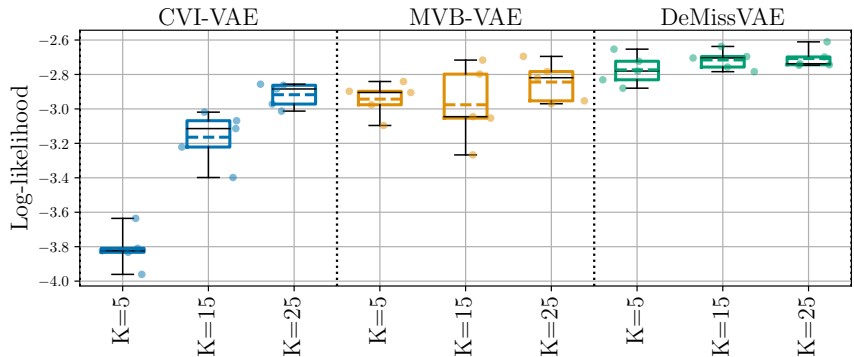

Figure 6: *A control study on a VAE model with 2D latent space (see additional details in appendix E.1), investigating the importance of the two-objective approach in DeMissVAE (green) and two control methods (blue and yellow).* In CVI-VAE (blue) we fit both the encoder and decoder using eq. (10), and in MVB-VAE (yellow) we fit both the encoder and decoder using eq. (12). The log-likelihood is computed on a held-out test data set by numerically integrating the 2D latent space of the VAE.

---

**Algorithm 1** Shared computation of the DeMissVAE learning objectives

**Input:** parameters $\boldsymbol{\theta}$ and $\boldsymbol{\phi}$, number of latent samples $L$, completed data-point $(\boldsymbol{x}_{\text{obs}}^i, \boldsymbol{x}_{\text{mis}}^{ik})$
1: $\boldsymbol{\psi}^{ik} \leftarrow \texttt{Encoder}(\boldsymbol{x}_{\text{obs}}^i, \boldsymbol{x}_{\text{mis}}^{ik}; \boldsymbol{\phi})$ $\qquad\qquad$ ▷ Compute parameters of the variational distribution
2: $\boldsymbol{z}_1, \dots, \boldsymbol{z}_L \sim q(\boldsymbol{z} \mid \boldsymbol{x}_{\text{obs}}^i, \boldsymbol{x}_{\text{mis}}^{ik}; \boldsymbol{\psi}^{ik})$ $\qquad\qquad\qquad\qquad$ ▷ Sample latents $\boldsymbol{z}$
3: $\boldsymbol{\eta}_l \leftarrow \texttt{Decoder}(\boldsymbol{z}_l; \boldsymbol{\theta})$ for $\forall l \in [1, L]$ $\qquad\qquad$ ▷ Compute parameters of the generative distribution
4: **def** $\mathcal{L}_{\text{CVI}}^{\boldsymbol{\theta}}(\boldsymbol{z}_1, \dots, \boldsymbol{z}_L, \boldsymbol{\eta}_1, \dots, \boldsymbol{\eta}_L)$: $\qquad\qquad\qquad$ ▷ Procedure for estimating eq. (10)
5: $\qquad$ **return** $\frac{1}{L} \sum_{l=1}^L \log p(\boldsymbol{x}_{\text{obs}}^i, \boldsymbol{z}_l; \boldsymbol{\eta}_l)$
6: **def** $\mathcal{L}_{\text{LMVB}}^{\boldsymbol{\phi}}(\boldsymbol{\psi}^{ik}, \boldsymbol{z}_1, \dots, \boldsymbol{z}_L, \boldsymbol{\eta}_1, \dots, \boldsymbol{\eta}_L)$: $\qquad\qquad$ ▷ Procedure for estimating eq. (12)
7: $\qquad$ **return** $\frac{1}{L} \sum_{l=1}^L \log p(\boldsymbol{x}_{\text{obs}}^i, \boldsymbol{x}_{\text{mis}}^{ik}, \boldsymbol{z}_l; \boldsymbol{\eta}_l) - \log q(\boldsymbol{z}_l \mid \boldsymbol{x}_{\text{obs}}^i, \boldsymbol{x}_{\text{mis}}^{ik}; \boldsymbol{\psi}^{ik})$
**return** $\mathcal{L}_{\text{CVI}}^{\boldsymbol{\theta}}(\boldsymbol{z}_1, \dots, \boldsymbol{z}_L, \boldsymbol{\eta}_1, \dots, \boldsymbol{\eta}_L), \mathcal{L}_{\text{LMVB}}^{\boldsymbol{\phi}}(\boldsymbol{\psi}^{ik}, \boldsymbol{z}_1, \dots, \boldsymbol{z}_L, \boldsymbol{\eta}_1, \dots, \boldsymbol{\eta}_L)$

---

use cheap approximate iterative sampling methods such as pseudo-Gibbs (Rezende et al., 2014, Appendix F), Metropolis-within-Gibbs (MWG, Mattei & Frellsen, 2018a), or latent-adaptive importance resampling (LAIR, Simkus & Gutmann, 2023). Moreover, since the model and the variational distributions are initialised randomly, we skip the imputation step during the first epoch over the data.

2. **Parameter update.** Update the parameters using stochastic gradient ascent on $\mathcal{L}_{\text{CVI}}^{\boldsymbol{\theta}}$ and $\mathcal{L}_{\text{LMVB}}^{\boldsymbol{\phi}}$ in eqs. (10) and (12) with the imputations from $f^t(\boldsymbol{x}_{\text{mis}} \mid \boldsymbol{x}_{\text{obs}})$.

**Efficient parameter update.** While the two objectives for $p_{\boldsymbol{\theta}}$ and $q_{\boldsymbol{\phi}}$ in eqs. (10) and (12) are different, a major part of the computation can be shared, as shown in algorithm 1. As usual, the objectives are approximated using Monte Carlo averaging and require only one evaluation of the generative model, including the encoder, decoder, and prior, for each completed data-point $(\boldsymbol{x}_{\text{obs}}^i, \boldsymbol{x}_{\text{mis}}^{ik})$. Therefore, only backpropagation needs to be performed separately and the overall per-iteration computational cost of optimising the two objectives is about 1.67 times the cost of a fully-observed VAE optimisation (instead of 2 times if implemented naïvely).[10]

**Efficient imputation.** To make the imputation step efficient, the imputation distribution $f^t(\boldsymbol{x}_{\text{mis}} \mid \boldsymbol{x}_{\text{obs}})$ is "persistent" between iterations, that is, the imputation distribution from the previous iteration

---

[10]The cost of backpropagation is about 2 times the cost of a forward pass (Burda et al., 2015).

$f^{t-1}(\boldsymbol{x}_{\mathrm{mis}} \mid \boldsymbol{x}_{\mathrm{obs}})$ is used to initialise the iterative approximate VAE sampler at iteration $t$.[11] Moreover, an iteration of a pseudo-Gibbs, MWG, or LAIR samplers uses the same quantities as the objectives $\mathcal{L}^{\boldsymbol{\theta}}_{\mathrm{CVI}}$ and $\mathcal{L}^{\boldsymbol{\phi}}_{\mathrm{LMVB}}$ in eqs. (10) and (12), and hence the cost of one iteration of the sampler in the imputation step can be shared with the cost of computation of the learning objectives. However, it is important to note that the accuracy of imputations affects the accuracy of the estimated model, and hence better estimation can be achieved by increasing the computational budget for imputation or using better imputation methods.

---

[11] Persistent samplers have been used in the past to increase efficiency of maximum-likelihood estimation methods (Younes, 1999; Tieleman, 2008; Simkus et al., 2023).

# E    Experiment details

In this appendix we provide additional details on the experiments.

## E.1    Mixture-of-Gaussians data with a 2D latent VAE

We generated a random 5D mixture-of-Gaussians model with 15 components by sampling the mixture covariance matrices from the inverse Wishart distribution $\mathcal{W}^{-1}(\nu = D, \Psi = \mathbf{I})$, means from the Gaussian distribution $\mathcal{N}(\mu = \mathbf{0}, \sigma = \mathbf{3})$ and the component probabilities from Dirichlet distribution $\mathrm{Dir}(\alpha = \mathbf{1})$ (uniform). The model was then standardised to have a zero mean and a standard deviation of one. The pairwise marginal densities of the generated distribution is visualised in fig. 7 showing a highly-complex and multimodal distribution, and the generated parameters and data used in this paper are available in the shared code repository. We simulated a 20K sample data set used to fit the VAEs.

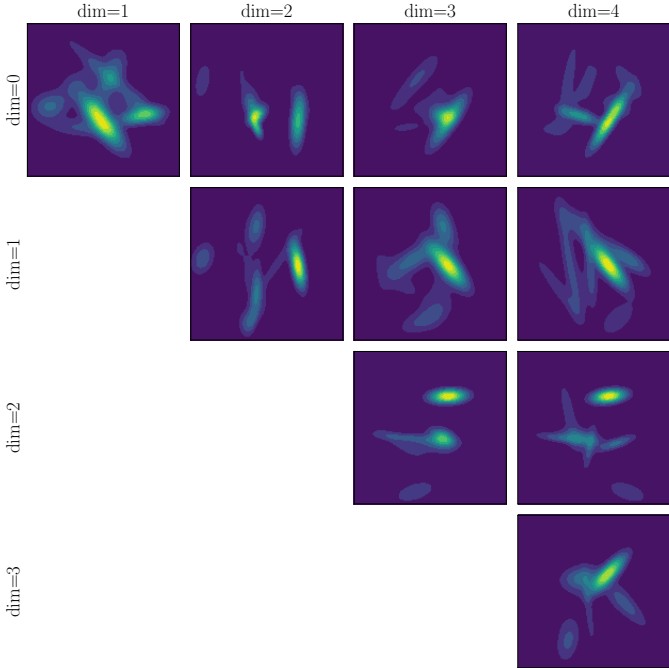

Figure 7: The pairwise marginals of the ground-truth Mixture-of-Gaussians distribution.

We then fitted a VAE model with 2-dimensional latent space using diagonal Gaussian encoder and decoder distributions, and a fixed standard Normal prior. For the decoder and encoder networks we used fully-connected residual neural networks with 3 residual blocks, 200 hidden dimensions, and ReLU activations. To optimise the model parameters we have used AMSGrad optimiser (Reddi et al., 2018) with a learning rate of $10^{-3}$ for a total of 500 epochs.

The hyperameters are listed in table 3, note that the total number of samples was the same for all methods (i.e. 5/15/25). Moreover, we have used "sticking-the-landing" (STL) gradients (Roeder et al., 2017) to reduce gradient variance for all methods.[12]

---

[12]We have also evaluated the doubly-reparametrised gradients (DReG, Tucker et al., 2018) for IWAE methods but found STL to perform similar or better.

| Method | $Z$ | $K$ | $I$ | Mixture sampling |
|--------|-----|-----|-----|------------------|
| MVAE | 5/15/25 | 1 | — | — |
| MissVAE | 5/15/25 | 5/15/25 | — | Ancestral |
| MissSVAE | 1 | 5/15/25 | — | Stratified |
| MIWAE | 1 | 1 | 5/15/25 | — |
| MissIWAE | 1 | 5/15/25 | 5/15/25 | Ancestral |
| MissSIWAE | 1 | 5/15/25 | 1 | Stratified |
| DeMissVAE | 1 | 5/15/25 | — | LAIR (1 iteration, $R = 0$) (Simkus & Gutmann, 2023) |

Table 3: *Method hyperparameters on MoG data.*

## E.2 UCI data sets

We fit the VAEs on four data sets from the UCI repository (Dua & Graff, 2017) with the preprocessing of (Papamakarios et al., 2017). The VAE model uses diagonal Gaussian encoder and decoder distributions regularised such that the standard deviation $\geq 10^{-5}$ (Mattei & Frellsen, 2018a), and a fixed standard Normal prior. The latent space is 16-dimensional, except for the MINIBOONE where 32 dimensions were used.

The encoder and decoder networks used fully-connected residual neural networks with 2 residual blocks (except for on the MINIBOONE dataset where 5 blocks were used in the encoder) with 256 hidden dimensionality, and ReLU activations. A dropout of 0.5 was used on the MINIBOONE dataset. The parameters were optimised using AMSGrad optimiser (Reddi et al., 2018) with a learning rate of $10^{-3}$ and cosine learning rate schedule for a total of 200K iterations (or 22K iterations on MINIBOONE). As before, STL gradients (Roeder et al., 2017) were used to reduce the variance for all methods. DeMissVAE used the LAIR sampler (Simkus & Gutmann, 2023) with $K = 5$ $R = 1$ and 10 iterations. Moreover we have used gradient norm clipping to stabilise DeMissVAE with the maximum norm set to 1 (except for POWER dataset where we set it to 0.5).

## E.3 MNIST and Omniglot data sets

We fit a VAE on statically binarised MNIST and Omniglot data sets (Lake et al., 2015) downsampled to 28x28 pixels. The VAE uses diagonal Gaussian decoder distributions regularised such that the standard deviation $\geq 10^{-5}$ (Mattei & Frellsen, 2018a), a fixed standard Normal prior, and a Bernoulli decoder distribution. The latent space is 50-dimensional.

For both MNIST and Omniglot we have used convolutional ResNet neural networks for the encoder and decoder with 4 residual blocks, ReLU activations, and dropout probability of 0.3. For MNIST, the encoder the residual block hidden dimensionalities were 32, 64, 128, 256, and for the decoder they were 128,64,32,32. For Omniglot, the encoder the residual block hidden dimensionalities were 64, 128, 256, 512, and for the decoder they were 256,128,64,64. We used AMSGrad optimiser (Reddi et al., 2018) with $10^{-4}$ learning rate, cosine learning rate schedule, and STL gradients (Roeder et al., 2017) for 500 epochs for MNIST and 200 epochs for Omniglot.

For MVAE, we use 5 latent samples and for MIWAE we use 5 importance samples. For MissVAE we use $K = 5$ mixture components and sample 5 latent samples. For MissSVAE we use $K = 5$ mixture components and sample 1 sample from each component, for a total of 5 samples. For MissIWAE we use $K = 5$ components and sample 5 importance samples. for MissSIWAE we use $K = 5$ components and sample 1 sample from each component. For DeMissVAE we use $K = 5$ imputations and update them using a single step of pseudo-Gibbs (Rezende et al., 2014).

# F   Additional figures

In this appendix we provide additional figures for the experiments in this paper.

## F.1   Mixture-of-Gaussians data with a 2D latent VAE

In this section we show additional analysis on the mixture-of-Gaussians data, supplementing the results in section 6.1.

### F.1.1   Analysis of gradient variance with ancestral and stratified sampling

In section 6.1 we observed that the model estimation performance can depend on whether ancestral sampling (with implicit reparametrisation) or stratified sampling is used to approximate the expectations in eqs. (4), (5), (7) and (8), corresponding to MissVAE/MissIWAE and MissSVAE/MissSIWAE, respectively.

We analyse the signal-to-noise ratio (SNR) of the gradients w.r.t. $\phi$ and $\theta$ for the two approaches, which is defined as follows (Rainforth et al., 2019)

$$\text{SNR}(\phi) = \left| \frac{\mathbb{E}\left[\Delta(\phi)\right]}{\sigma[\Delta(\phi)]} \right|, \quad \text{and} \quad \text{SNR}(\theta) = \left| \frac{\mathbb{E}\left[\Delta(\theta)\right]}{\sigma[\Delta(\theta)]} \right|,$$

where $\Delta(\cdot)$ denotes the gradient estimate, and $\sigma[\cdot]$ is the standard deviation of a random variable. We estimate the SNR by computing the expectation and standard deviation over the entire training epoch.

The SNR for $\phi$ and $\theta$ is plotted in fig. 8. We observe that the stratified approaches (MissSVAE and MissSIWAE) generally have higher SNR. This is possibly the reason why MissSVAE and MissSIWAE have achieved better model accuracy than the ancestral approaches (MissVAE and MissIWAE) in section 6.1.

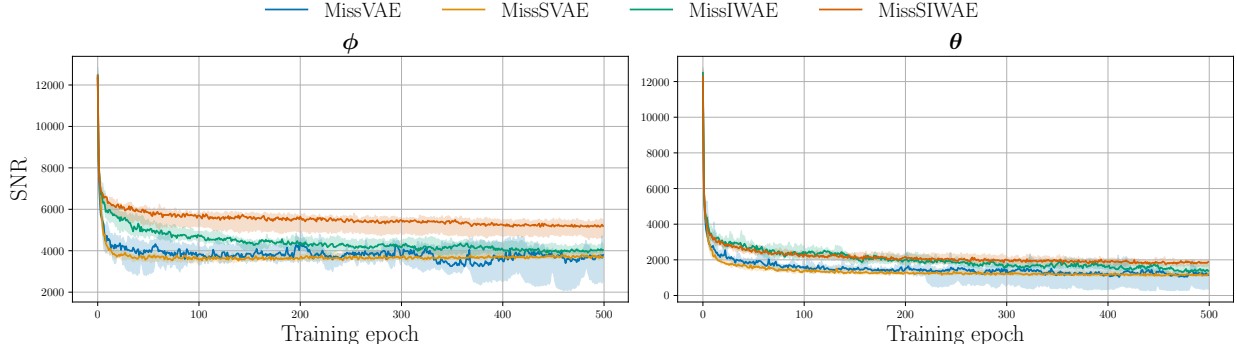

Figure 8: *Signal-to-noise ratio (SNR, higher is better) of the gradients w.r.t. encoder parameters $\phi$ (left) and decoder parameters $\theta$ (right).* For all methods we used a budget of 5 samples from the variational distribution (see appendix E.1 for more details). We show the median SNR over 5 independent runs and a 90% confidence interval.

### F.1.2   Analysis of the model posteriors

In figs. 9 to 11 we visualise the model posteriors with complete and incomplete data, $p_\theta(z \mid x)$ and $p_\theta(z \mid x_{\text{obs}})$, respectively, and the variational distribution $q_\phi(z \mid \cdot)$ that was used to fit the model via the variational ELBO. For each method we have used a budget of 25 samples from the variational distribution during training (additional details are in appendix E.1). Each figure shows the posteriors for 5 training data-points using distinct colours.

Figure 9 shows MVAE, MissVAE, and MissSVAE model posteriors $p_\theta(z \mid x)$ and $p_\theta(z \mid x_{\text{obs}})$, as well as the variational distribution $q_\phi(z \mid x_{\text{obs}})$, which approximates the incomplete-data posterior. As motivated

in section 3 we observe that the Gaussian posterior in MVAE (first row) is not sufficiently flexible to approximate the complex incomplete-data posteriors $p_{\boldsymbol{\theta}}(\boldsymbol{z} \mid \boldsymbol{x}_{\mathrm{obs}})$. On the other hand, the mixture-variational approaches, MissVAE (second row) and MissSVAE (third row), are able to well-approximate the incomplete-data posteriors.

Figure 10 shows MIWAE, MissIWAE, and MissSIWAE model posteriors $p_{\boldsymbol{\theta}}(\boldsymbol{z} \mid \boldsymbol{x})$ and $p_{\boldsymbol{\theta}}(\boldsymbol{z} \mid \boldsymbol{x}_{\mathrm{obs}})$, as well as the variational proposal $q_{\boldsymbol{\phi}}(\boldsymbol{z} \mid \boldsymbol{x}_{\mathrm{obs}})$ and the importance-weighted semi-implicit distribution $q_{\boldsymbol{\phi}, \boldsymbol{\theta}, I=25}^{\mathrm{IW}}(\boldsymbol{z} \mid \boldsymbol{x}_{\mathrm{obs}})$ that arises from sampling the variational proposal $q_{\boldsymbol{\phi}}(\boldsymbol{z} \mid \boldsymbol{x}_{\mathrm{obs}})$ and re-sampling using importance weights $w(\boldsymbol{z}) = p_{\boldsymbol{\theta}}(\boldsymbol{x}_{\mathrm{obs}}, \boldsymbol{z})/q_{\boldsymbol{\phi}}(\boldsymbol{z} \mid \boldsymbol{x}_{\mathrm{obs}})$ (Cremer et al., 2017). Similar to the MVAE case above, the variational proposal $q_{\boldsymbol{\phi}}(\boldsymbol{z} \mid \boldsymbol{x}_{\mathrm{obs}})$ in MIWAE (first row) is quite far from the model posterior $p_{\boldsymbol{\theta}}(\boldsymbol{z} \mid \boldsymbol{x}_{\mathrm{obs}})$, but the importance-weighted bound in eq. (7) is able to re-weigh the samples to sufficiently-well match the model posterior, as shown in the fourth column. However, as efficiency of importance sampling depends on the discrepancy between the proposal and the target distributions, we can expect that more flexible variational distributions may improve the performance of the importance-weighted ELBO methods. Importantly, we show that the variational-mixture approaches, MissIWAE (second row) and MissSIWAE (third row), are able to adapt the variational proposals to the incomplete-data posteriors well, and as a result achieve better efficiency than MIWAE.

Figure 11 shows DeMissVAE model posteriors $p_{\boldsymbol{\theta}}(\boldsymbol{z} \mid \boldsymbol{x})$ and $p_{\boldsymbol{\theta}}(\boldsymbol{z} \mid \boldsymbol{x}_{\mathrm{obs}})$, the variational distribution $q_{\boldsymbol{\phi}}(\boldsymbol{z} \mid \boldsymbol{x})$, which approximates the *complete*-data posterior, and the imputation-mixture $q_{\boldsymbol{\phi}, f^t}(\boldsymbol{z} \mid \boldsymbol{x}_{\mathrm{obs}})$ approximated using the 25 imputations in $f^t(\boldsymbol{x}_{\mathrm{mis}} \mid \boldsymbol{x}_{\mathrm{obs}})$ at the end of training. We observe similar behaviour to fig. 1, where the complete data posteriors $p_{\boldsymbol{\theta}}(\boldsymbol{z} \mid \boldsymbol{x})$ are close to Gaussian but the incomplete-data posteriors $p_{\boldsymbol{\theta}}(\boldsymbol{z} \mid \boldsymbol{x}_{\mathrm{obs}})$ are irregular. As we show in section 6.1, DeMissVAE is capable of fitting the model well by learning the completed-data variational distribution $q_{\boldsymbol{\phi}}(\boldsymbol{z} \mid \boldsymbol{x})$ (third column) and using the imputation-mixture in eq. (9) to approximate the incomplete data posterior $p_{\boldsymbol{\theta}}(\boldsymbol{z} \mid \boldsymbol{x}_{\mathrm{obs}})$. Moreover, we observe that the imputation-mixture $q_{\boldsymbol{\phi}, f^t}(\boldsymbol{z} \mid \boldsymbol{x}_{\mathrm{obs}})$ (fourth column) captures only one of the modes of the model posterior $p_{\boldsymbol{\theta}}(\boldsymbol{z} \mid \boldsymbol{x}_{\mathrm{obs}})$. This is a result of using a small imputation sampling budget, that is, using only a single iteration of LAIR to update the imputations (see more details in appendix D), and hence better accuracy can be achieved by trading-off some computational cost to obtain better imputations that would ensure a better representation of the imputation distribution. Nonetheless, as observed in fig. 2, DeMissVAE achieves good model accuracy despite potentially sub-optimal imputations, further signifying the importance of the two learning objectives for the encoder and decoder distributions in section 4.2 and appendix C.

Interestingly, by comparing the complete-data posteriors $p_{\boldsymbol{\theta}}(\boldsymbol{z} \mid \boldsymbol{x})$ (first column) in figs. 9 to 11, we observe that they are slightly more irregular than in the complete case in fig. 1, except for DeMissVAE whose posteriors are nearly Gaussian. The irregularity is stronger for the importance-weighted ELBO-based methods in fig. 10. This is in line with the observation by Burda et al. (2015, Appendix C) and Cremer et al. (2018, Section 5.4) that VAEs trained with more flexible variational distributions tend to learn a more complex model posterior. This means that using the importance-weighted bounds, and to a lesser extent the finite variational-mixture approaches from section 4.1, to fit VAEs on incomplete data may result in worse-structured latent spaces, compared to models fitted on complete data. On the other hand, we observe that DeMissVAE learns a better-structured latent space, with the posterior $p_{\boldsymbol{\theta}}(\boldsymbol{z} \mid \boldsymbol{x})$ close to a Gaussian, that is comparable to the complete case. This suggests that the decomposed approach in DeMissVAE may be important in cases where the latent space needs to be regular, at the additional cost of obtaining missing data imputations (see appendix D).

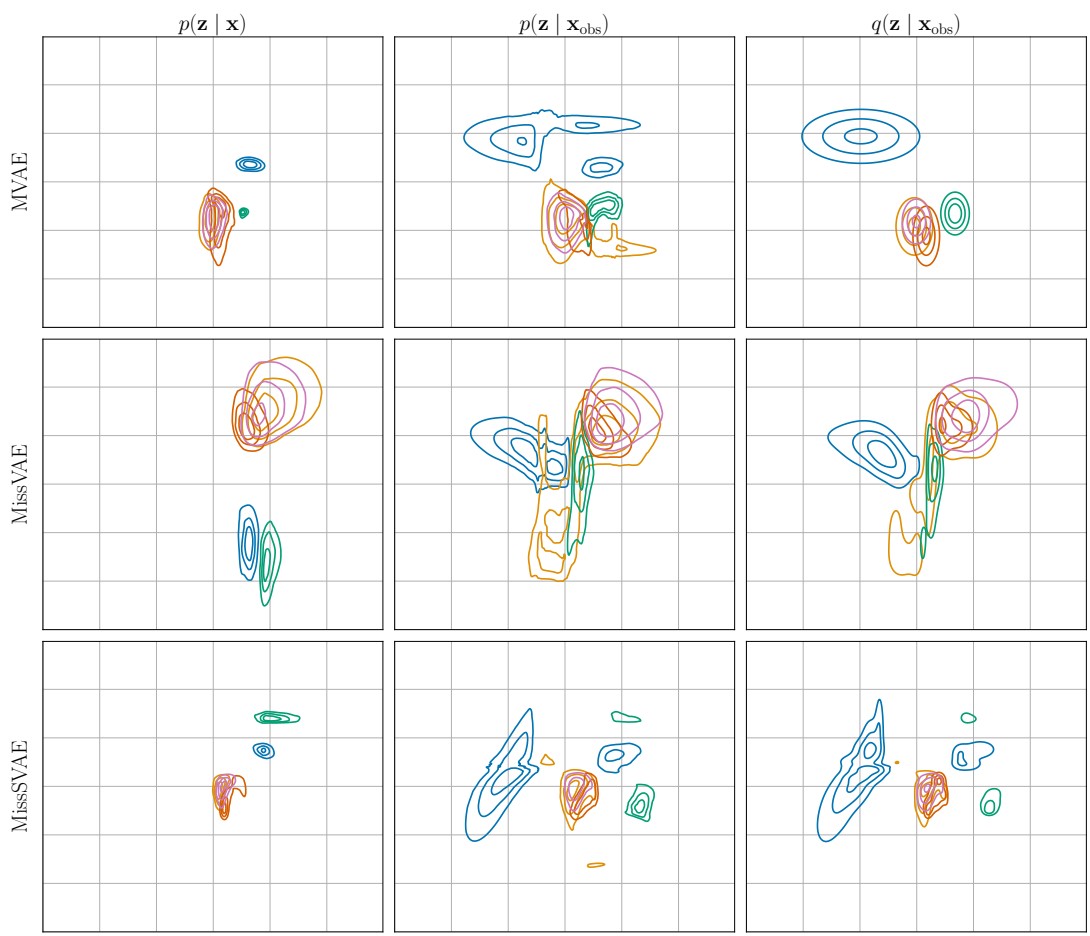

Figure 9: *Posterior distributions of MVAE, MissVAE, and MissSVAE.* First column: the model posterior $p_{\boldsymbol{\theta}}(\boldsymbol{z} \mid \boldsymbol{x})$ under complete data $\boldsymbol{x}$. Second column: the model posterior $p_{\boldsymbol{\theta}}(\boldsymbol{z} \mid \boldsymbol{x}_{\mathrm{obs}})$ under incomplete data $\boldsymbol{x}_{\mathrm{obs}}$. Third column: variational approximation $q_{\boldsymbol{\phi}}(\boldsymbol{z} \mid \boldsymbol{x}_{\mathrm{obs}})$ of the incomplete posterior $p_{\boldsymbol{\theta}}(\boldsymbol{z} \mid \boldsymbol{x}_{\mathrm{obs}})$ obtained at the end of training.

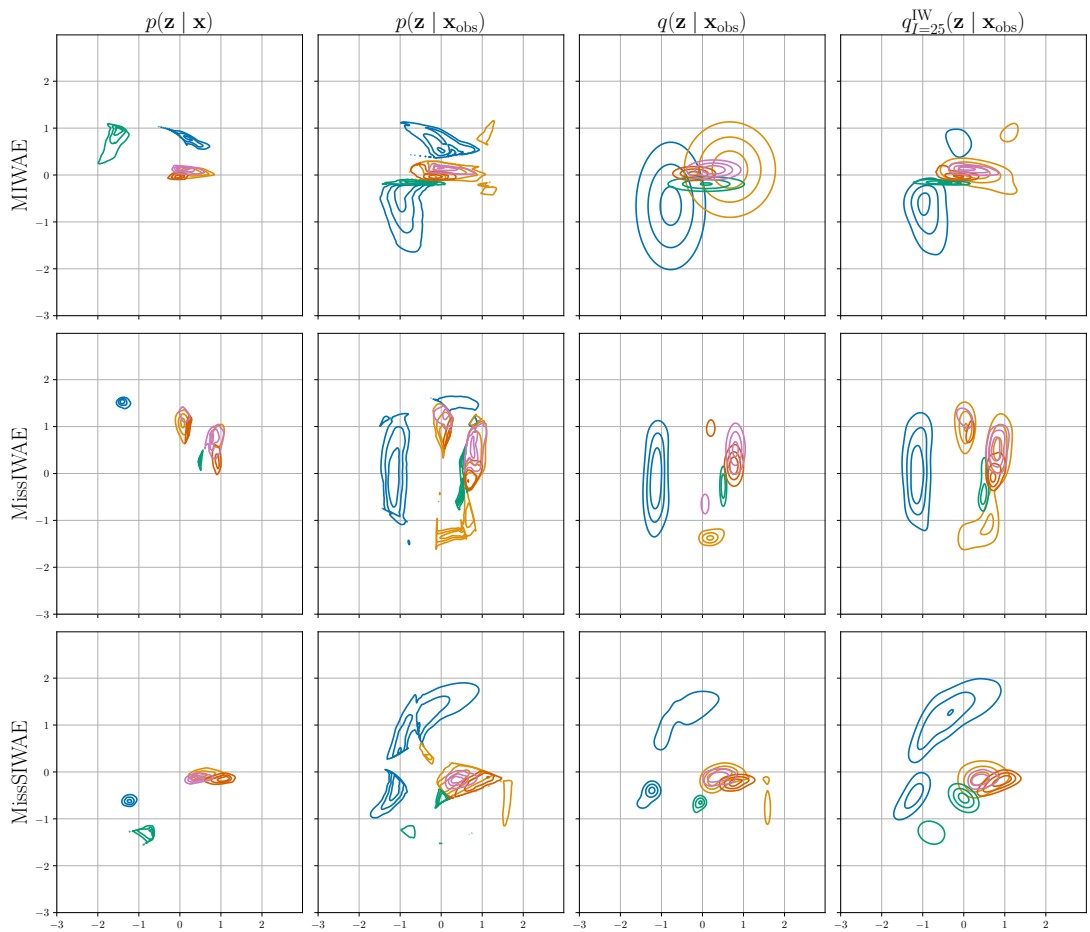

Figure 10: *Posterior distributions of MIWAE, MissIWAE, and MissSIWAE.* First column: the model posterior $p_{\boldsymbol{\theta}}(\boldsymbol{z} \mid \boldsymbol{x})$ under complete data $\boldsymbol{x}$. Second column: the model posterior $p_{\boldsymbol{\theta}}(\boldsymbol{z} \mid \boldsymbol{x}_{\mathrm{obs}})$ under incomplete data $\boldsymbol{x}_{\mathrm{obs}}$. Third column: variational proposal $q_{\boldsymbol{\phi}}(\boldsymbol{z} \mid \boldsymbol{x}_{\mathrm{obs}})$ for an incomplete data-point $\boldsymbol{x}_{\mathrm{obs}}$ obtained at the end of training. Fourth column: importance-weighted variational distribution $q_{\boldsymbol{\phi}}^{\mathrm{IW}}{}_{I=25}(\boldsymbol{z} \mid \boldsymbol{x}_{\mathrm{obs}})$ for an incomplete data-point $\boldsymbol{x}_{\mathrm{obs}}$ obtained after re-weighting samples from $q_{\boldsymbol{\phi}}(\boldsymbol{z} \mid \boldsymbol{x}_{\mathrm{obs}})$ (Cremer et al., 2017).

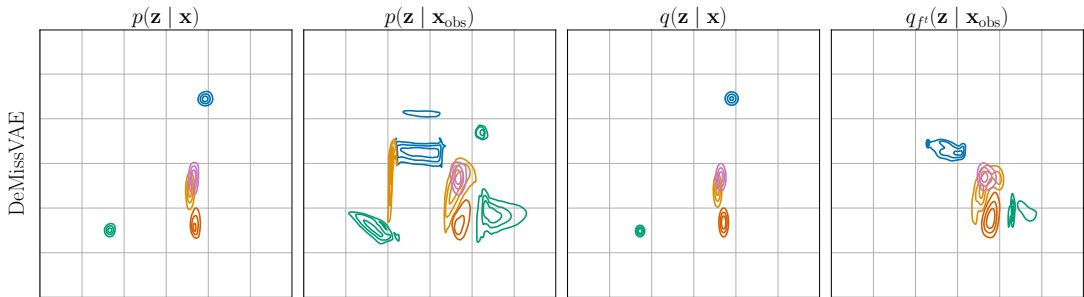

Figure 11: *Posterior distributions of DeMissVAE.* First: the model posterior $p_{\boldsymbol{\theta}}(\boldsymbol{z} \mid \boldsymbol{x})$ under complete data $\boldsymbol{x}$. Second: the model posterior $p_{\boldsymbol{\theta}}(\boldsymbol{z} \mid \boldsymbol{x}_{\mathrm{obs}})$ under incomplete data $\boldsymbol{x}_{\mathrm{obs}}$. Third: variational approximation $q_{\boldsymbol{\phi}}(\boldsymbol{z} \mid \boldsymbol{x})$ of the complete-data posterior $p_{\boldsymbol{\theta}}(\boldsymbol{z} \mid \boldsymbol{x})$ obtained at the end of training. Fourth: the variational imputation-mixture distribution in eq. (9) using the imputation distribution $f^t(\boldsymbol{x}_{\mathrm{mis}} \mid \boldsymbol{x}_{\mathrm{obs}})$ obtained at the end of training, approximated using a Monte Carlo average with 25 imputations.

## F.2   UCI data sets

In fig. 12 we plot the Fréchet inception distance (FID, Heusel et al., 2017) versus training iteration on the UCI datasets. The results closely mimic the log-likelihood results in section 6.2. Importantly, we observe that using mixture variational distributions becomes more important as the missingness fraction increases, causing the posterior distributions to be more complex.

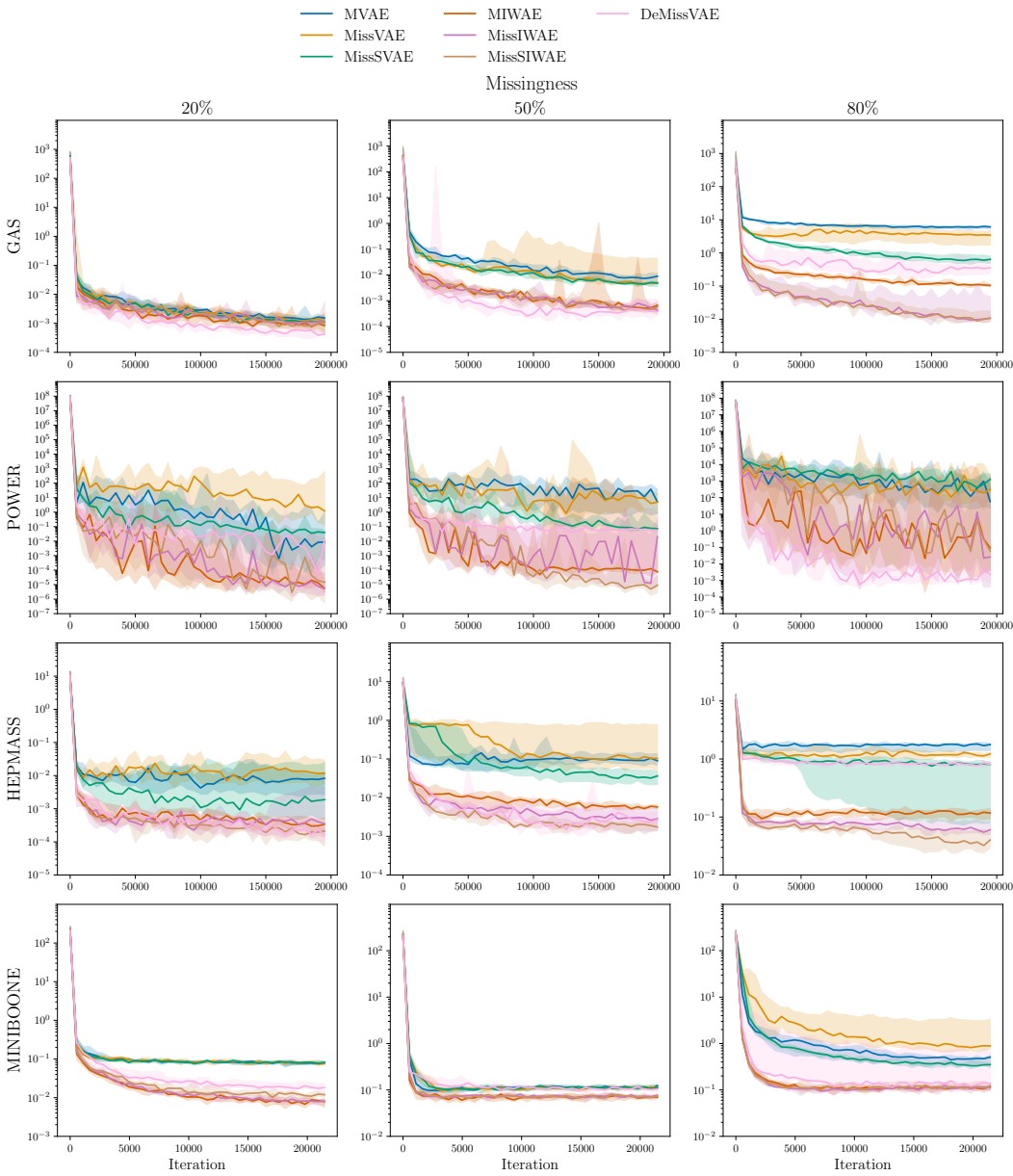

Figure 12: *FID (lower is better) between the model and the complete test data versus training iterations.* The FID is computed using features of the last encoder layer of an independent VAE model trained on complete data. Lines show the median of 5 independent runs and the intervals show 90% confidence.

### F.3 MNIST and Omniglot data sets, latent dimensionality

Rather than handling the posterior complexity due to missingness with variational mixtures, an alternative approach may be to increase the capacity of the model by making the latent space dimensionality larger. In fig. 13 we plot the estimated test log-likelihood against latent variable dimensionality.

On MNIST data, the effect from increasing the latent space is small, with the exception of DeMissVAE. The reason why DeMissVAE improves quite significantly with latent space dimensionality may be more due to easier sampling than a larger capacity of the model. This again highlights that DeMissVAE, when used with efficient sampling methods, can be an efficient approach to handle data missingness.

On Omniglot data, we observe a small improvement for MIWAE, MVAE, and the stratified MissSIWAE, while the other methods either remained at about the same accuracy or declined. However, there is no significant change from the results in section 6.3.

Hence, while increasing latent dimensionality generally increases the capacity of the model, enabling the modelling of more complex distributions, overall we observe that it provides minimal effect for dealing with data missingness.

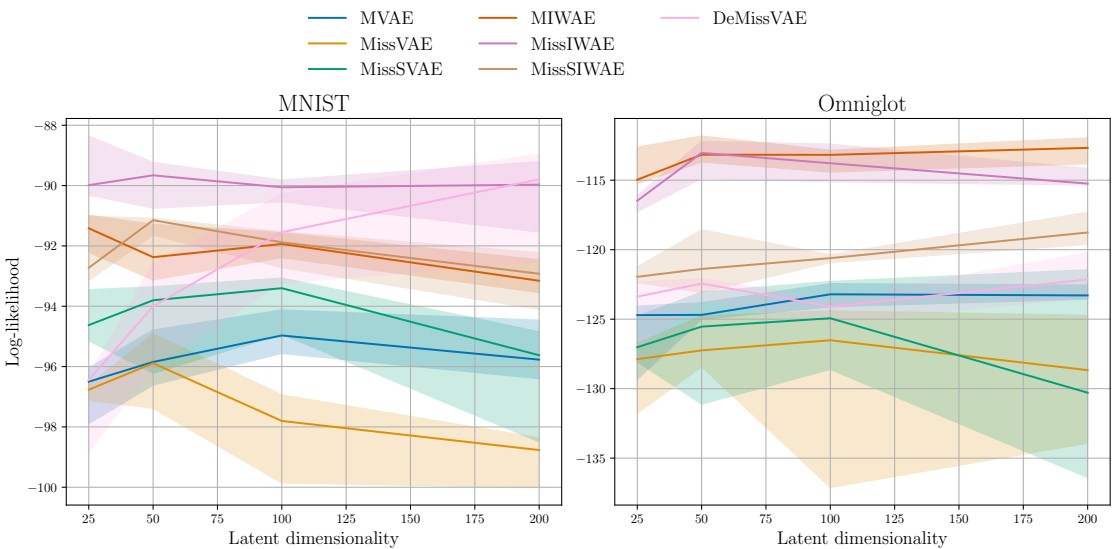

Figure 13: *Estimate of the test log-likelihood against latent dimensionality on MNIST and Omniglot data sets.* The log-likelihood was estimated on complete test data using the IWELBO with $I = 1000$. The curves show median performance over 5 independent runs of the methods and the intervals show the 90% centered interval.

