# OpenReview forum: "Improving Variational Autoencoder Estimation from Incomplete Data with Mixture Variational Families"
_TMLR — Accepted by TMLR_

### Review · Reviewer_WraX · 2024-03-12

**Summary Of Contributions:**

The paper studies variational inference with a VAE when parts of the observations are missing at random. They point out that this can make the correct form of the posterior more complex, necessitating more suitable variational forms. They consider mixtures of variational posteriors and posteriors which first estimate and sample imputations before doing standard variational inference.

**Audience:**

Yes

**Claims And Evidence:**

Yes

**Requested Changes:**

See above

**Strengths And Weaknesses:**

**Overall I think the paper is worth accepting to TMLR and that some readers may find it useful (ideally with some clarifications).** The method itself is not new (using a mixture of variational posteriors, data imputation before inference) but the specific discussion of the difficulty of variational inference with missing data is interesting. In particular, it raises questions as to the appropriate way to handle missingness, and whether it is sufficient to just "scale up the posterior family" or if additional structure can be leveraged---particularly when the form of such structure may be available to domain experts ahead of time.

**My two main suggestions are to improve the clarity of the writing at certain points and to expand on the somewhat limited experiments.**

## Regarding writing clarity, there are quite a few segments that could be made much easier for readers to follow:

Section 2 should be much clearer about a lot of definitions, variable domains, etc. For example,
* In the first paragraph, what is $D$? I guess it must be the data dimension? It looks like it's not defined until the end of the section.
* Is $m^i$ itself observed? The paper describes methods which feed $m^i$ as input, but then later in the paper describes the function $f_t$ as imputing conditioned on only $x^i_{obs}$, presumably without $m^i$.
* Similarly, how can it be that MAR is equivalent to $p(m^i \mid x^i) = p(m^i \mid x^i_{obs})$? Wouldn't $p(m^i \mid x^i_{obs})$ have zero mass on any $m^i$ with 0 in the dimensions of $x^i_{obs}$? If the $x^i_{obs}$ are the values in $x^i$ which are observed, then they can only be defined *after* $m^i$ is chosen, no? This confusion could be addressed by more precisely specifying these definitions.

The setup for Figure 1 is not explained **at all**. What each of the plots depicts is clear, but how are these results generated?

* What are the domains/distributions of $x$ and $z$? What is the missingness distribution? Does it match the "mixture" assumption made by the method in the rightmost plot? How would it look when it does not?
* At the beginning of Section 4.1 the authors write: “In section 3 we saw that a good approximation of the incomplete data posterior $p_\theta(z \mid x^i_{obs})$ would be the imputation-mixture”. Isn't this circular reasoning? The rightmost plot of Figure 1 is being used to motivate a new method... which is the method being depicted in Figure 1? Ultimately this seems to just be saying "using a variational posterior family which gets closer to the true posterior can do a better job", which is obvious...

Similarly to the first sentence of 4.1, the first sentence after Eq (9) is extremely speculative---there is *some* reasoning offered closer to the end of that section, these should be connected. **Throughout the paper, whenever a claim is made about what should or shouldn't work and why, it's up to the authors to offer justification.**

## Regarding the somewhat limited experiments

It's not too surprising that the existing MIWAE is equal/better in most cases, since as I mentioned the objectives discussed in this work don't do anything explicitly new, instead applying existing ideas to missingness (a worthwhile application!). However, I am somewhat disappointed that the experiments are so simplistic. There are a lot of ways this could be improved, off the top of my head:

* Why not use more complex real-world data, instead of, e.g., GAS/MNIST/Omniglot?
* Why are all the missingness patterns so basic (e.g. very often *explicitly* inducing mixture posteriors, such as randomly masking two quadrants of an image), rather than more interesting distributions?
* Why not evaluate in settings when the data is *not* MAR, to determine how important that assumption is to each of the methods?
* Does the relative ranking of these methods change if you use a more complex (i.e., larger) encoder/decoder? Explicitly structuring the posterior is of course helpful, but simply choosing a more flexible family could also potentially work. How do these trade off? (In particular, were the experiments on more complex data I would understand if this were computationally costly; but for such simple data, these types of experiments would still be pretty trivial).

Not all of these suggestions would need to be implemented, of course, but these experiments feel a bit like just checking boxes rather than truly exploring the interesting problem of variational inference under missingness. I think investigating a bit more deeply would make the paper much more informative overall.

---

> ### Author Response · Authors · 2024-04-12
> **Response (Part 1)**
>
> We thank the reviewer for the feedback and questions. We respond to each of the raised points one-by-one below. In the updated manuscript, the updated text is highlighted using red colour.
>
> > * In the first paragraph [of Section 2], what is $D$? I guess it must be the data dimension? It looks like it's not defined until the end of the section.
>
> Thank you for noting this, $D$ is the dimensionality of the data-points. We have updated the paragraph with this information.
>
> > * Is $m^i$ itself observed? The paper describes methods which feed $m^i$ as input, but then later in the paper describes the function $f^t$ as imputing conditioned on only $x_\text{obs}$, presumably without $m^i$.
>
> Yes, the missing dimensions are typically considered known from the data set. The reason for feeding the missingness mask $m^i$ to the neural network in some deep learning methods, is that it helps the network to work out which input dimensions are observed and which are not. Under the MAR assumption, which we adopt in the paper, statistical inference of the parameters $\theta$ and the imputation distribution of the missing values do not depend on $m^i$. However, to be able to actually impute the missing values, we do need to know which dimensions are missing (i.e. the value of $m^i$)! We chose to not make this dependency on $m^i$ explicit in our notation for $f^t$ since the distribution itself does not depend on $m^i$ even though the target dimensions which are imputed using $f^t$ are defined by $m^i$.
>
> In Section 2 of the updated manuscript, we have added references to the appropriate literature that explains when ignoring the missingness indicator is statistically valid.
>
> > * Similarly, how can it be that MAR is equivalent to $p(m^i \mid x^i) = p(m^i \mid x_{\text{obs}}^i)$? Wouldn't $p(m^i \mid x_{\text{obs}}^i)$ have zero mass on any $m^i$ with 0 in the dimensions of $x_{\text{obs}}$? If the $x_{\text{obs}}^i$ are the values in $x^i$ which are observed, then they can only be defined _after_ $m^i$ is chosen, no? This confusion could be addressed by more precisely specifying these definitions.
>
> In the paper we adopted a common approach to handle observed and missing variables that uses the missingness mask $m$ as an additional (but observed) random variable. The missingness mask $m$ indicates which dimensions/indices are observed or missing while $x_{\text{obs}}$ and $x_{\text{mis}}$ denote the values of the dimensions/indices. Given $x_{\text{obs}}$ alone, we do not know which dimensions/indices are actually observed; that information is provided by $m$. This is why $p(m^i \mid x_{\text{obs}}^i)$ is not a degenerate distribution. Moreover, using the Python-inspired notation $x_{\text{obs}} = x[m]$ and $x_{\text{mis}} = x[1-m]$, where $x[m]$ extracts the observed dimensions from $x$, and $x[1-m]$ extracts the missing dimensions, we can write the MAR assumption as follows: The data are MAR if $p(m \mid x) = p(m \mid \tilde x)$ for all $\forall m, x, \tilde x$, such that $x[m] = \tilde x[m]$ (in fact in the paper we adopt an even weaker MAR assumption known as _realised_ MAR).
>
> In the updated manuscript we have expanded on the notation, removed the equation $p(m^i \mid x^i) = p(m^i \mid x_{\text{obs}}^i)$ as it is actually unnecessary for the paper and may be confusing, and provided an additional reference to [1], which dissects the common MAR assumptions and their differences.
>
> > The setup for Figure 1 is not explained **at all**. What each of the plots depicts is clear, but how are these results generated?
> >
> > * What are the domains/distributions of $x$ and $z$? What is the missingness distribution? Does it match the "mixture" assumption made by the method in the rightmost plot? How would it look when it does not?
>
> The details about the model (including distributions of $x$ and $z$) used in Figure 1 were provided in the original manuscript in Footnote 3 (now Footnote 4). We understand that this can get easily overlooked. In the updated manuscript, the model and missingness details are thus now included in the caption. In particular, there is no "mixture" assumption in the missingness process, the mixture arises from the fact the incomplete data posterior $p(z\mid x_{\text{obs}})$ can be rewritten as an expectation $\mathbb{E}\_{p(x_{\text{mis}} \mid x_{\text{obs}})}[p(z \mid x_{\text{obs}}, x_{\text{mis}})]$, which can be viewed as a mixture over $p(x_{\text{mis}} \mid x_{\text{obs}})$.
>
> In the updated manuscript, we have rephrased the first part of paragraph 3 in Section 3 to clarify this.

---

> > ### Author Response · Authors · 2024-04-12
> > **Response (Part 2)**
> >
> > > * At the beginning of Section 4.1 the authors write: “In section 3 we saw that a good approximation of the incomplete data posterior $p_{\theta}(z \mid x_{\text{obs}})$ would be the imputation-mixture”. Isn't this circular reasoning? The rightmost plot of Figure 1 is being used to motivate a new method... which is the method being depicted in Figure 1? Ultimately this seems to just be saying "using a variational posterior family which gets closer to the true posterior can do a better job", which is obvious...
> >
> > Figure 1 does not depict the new method but builds motivation for it and illustrates the basic idea behind it; there is no circular reasoning. The motivation comes from the fact that the incomplete-data model posterior $p(z \mid x_{\text{obs}})$ can be viewed as a mixture $\mathbb{E}\_{p(x_{\text{mis}} \mid x_{\text{obs}})}[p(z \mid x_{\text{obs}}, x_{\text{mis}})]$. By replacing $p(z \mid x_{\text{obs}}, x_{\text{mis}})$ with $q(z \mid x_{\text{obs}}, x_{\text{mis}})$ we obtain the mixture approximation in the right side of Figure 1. Importantly, this motivation suggests that we can work with the same variational family in both fully-observed and incomplete data scenarios if we adopt a mixture approach, allowing us to carry inductive biases from the complete to the incomplete case, as explained in paragraph 3 of Section 3.
> >
> > We updated the text in paragraph 3 of Section 3 to better explain this point.
> >
> > > Similarly to the first sentence of 4.1, the first sentence after Eq (9) is extremely speculative---there is _some_ reasoning offered closer to the end of that section, these should be connected.
> >
> > Similar to the previous comment, the motivation comes from the simple decomposition of model posterior $p(z \mid x_{\text{obs}}) = \mathbb{E}\_{p(x_{\text{mis}} \mid x_{\text{obs}})}[p(z \mid x_{\text{obs}}, x_{\text{mis}})]$. Moreover, the explanations after Eq (13) provide a more technical justification.
> >
> > In the updated manuscript, we have updated the text after Eq (9) to repeat the intuition from Section 3, and have added signposting to the more technical justification that follows later in the section.
> >
> > > Why not use more complex real-world data, instead of, e.g., GAS/MNIST/Omniglot?
> >
> > In the paper we have included 6 real data sets and 1 synthetic data set, that are used to evaluate the 7 methods on different data modalities and models. The four UCI datasets, while low-dimensional, contain complex variable interdependencies, and as a result are often used for deep model benchmarking, for example in normalising flow papers [2]. MNIST and Omniglot are used to evaluate the methods on discrete data, such as images. Morevoer, for the four UCI datasets we have evaluated on 3 different missingness fractions. So in total there are $(4 \cdot 3 + 2) \cdot 7 = 98$ experiment runs (not including the synthetic data) that are all repeated 5 times to obtain uncertainty estimates. We believe that this evaluation sufficiently shows the usefulness of variational mixtures in incomplete data settings.
> >
> > > * Why are all the missingness patterns so basic (e.g. very often explicitly inducing mixture posteriors, such as randomly masking two quadrants of an image), rather than more interesting distributions?
> >
> > As explained in Appendix A, in images block-missingness is more _interesting_ (i.e. harder) than other types of missingness. For example, if the missingness was uniformly random, the incomplete-data posterior may not be too far from the complete data case. This is because pixels in an image contain a substantial amount of information about the neighbouring pixels, making many of them "redundant". On the other hand, masking blocks in an image, induces significant uncertainty, which is arguably more interesting.
> >
> > > * Why not evaluate in settings when the data is not MAR, to determine how important that assumption is to each of the methods?
> >
> > In this paper we have assumed MAR missingness from the start, a very common assumption in the literature. Imputation and estimation under MNAR (missing not at random) is more complicated: it additionally requires the estimation of the missingness distribution $p(m \mid x)$, imputation needs conditioning on $m$, _and_ the distributions $p(m \mid x)$ and $p(x)$ may not be identifiable from the data alone. Model estimation under MNAR is beyond the scope of this paper, and future work may build on this by incorporating the necessary additional assumptions and modifications that make estimation under MNAR feasible.

---

> > > ### Author Response · Authors · 2024-04-12
> > > **Response (Part 3)**
> > >
> > > > * Does the relative ranking of these methods change if you use a more complex (i.e., larger) encoder/decoder? Explicitly structuring the posterior is of course helpful, but simply choosing a more flexible family could also potentially work. How do these trade off? (In particular, were the experiments on more complex data I would understand if this were computationally costly; but for such simple data, these types of experiments would still be pretty trivial).
> > >
> > > In the updated manuscript we have included additional experiments in Appendix F.3, where we investigated the performance of the methods as we increase or reduce the latent variable dimensionality. This is motivated by the fact that increasing latent dimensionality can be seen to increase the flexibility of the model, and as a result be able to capture the complexities of incomplete data better. We found that the relative performance of the methods did not change.
> > >
> > >
> > > References:
> > >
> > > [1] Shaun Seaman, John Galati, Dan Jackson, and John Carlin. What Is Meant by “Missing at Random”?
> > >
> > > [2] George Papamakarios, Theo Pavlakou, and Iain Murray. Masked Autoregressive Flow for Density Estimation.

---

### Review · Reviewer_xSek · 2024-04-03

**Summary Of Contributions:**

The paper provides a methodological/numerical contribution to variational auto encoders under missing data. It provides 5 new methods based on mixtures and imputation methods. The new methods are justified heuristically in details. They are compared to two existing methods in a wide range of simulations. The related literature is carefully discussed.

**Audience:**

Yes

**Broader Impact Concerns:**

I do not have broader impact concerns.

**Claims And Evidence:**

Yes

**Requested Changes:**

Perhaps developing the benefit compared to MIWAE.

On page 3, "(2) may not minimized". Could it be phrased as "(2) cannot be set to zero"? It seems any function, such as (2), can be minimised with respect to its parameters, even if the minimal value is not zero?

**Strengths And Weaknesses:**

STRENGTHS: Many modern methods are discussed and analysed. In simulations, high dimensional problems can be addressed. The paper seems carefully written, and I did not find typos.

WEAKNESSES: The paper is numerically-oriented. There are no long proofs or theoretical guarantees. In my opinion this is not really a problem, since a paper can be numerically-oriented in essence.
In the various figures, the performance benefit compared to the baseline MIWAE is not clear (Figs 2 and 3).

---

> ### Author Response · Authors · 2024-04-12
> **Response**
>
> Thank you for your feedback. We respond to the raised points below. In the updated manuscript, the updated text is highlighted using red colour.
>
> > Perhaps developing the benefit compared to MIWAE.
>
> Indeed, our results show small improvement by using variational mixtures with the IWELBO bounds. This improvement is generally more pronounced for high missingness fractions (Figure 3), and on MNIST data (Figure 4).
> As we have discussed in Section 6.2, the reason why we see a relatively small improvement with the IWELBO bounds compared to ELBO bounds is that IWELBO can be viewed to be using a semi-implicitly defined variational distribution [1], which is able to mitigate the effects of missing data well. Hence, the MIWAE baseline, which uses IWELBO is a strong baseline to beat. An additional factor that may partially contribute to this, is that the IWELBO with mixture distribution may lead to mode collapse of the variational mixture as reported by [2], hence leading to a similar performance of MIWAE baseline that uses non-mixture distributions. Alternative importance-weighted bound formulations could be useful to mitigate this and is left as a future work.
>
> In the updated manuscript, we updated our discussion in Section 7 with the above points.
>
> > On page 3, "(2) may not minimized". Could it be phrased as "(2) cannot be set to zero"? It seems any function, such as (2), can be minimised with respect to its parameters, even if the minimal value is not zero?
>
> Thank you. We have corrected this to "may not be set to zero" in the updated manuscript.
>
> References:
>
> [1] Chris Cremer, Quaid Morris, and David Duvenaud. Reinterpreting Importance-Weighted Autoencoders.
>
> [2] Yuge Shi, N. Siddharth, Brooks Paige, and Philip Torr. Variational Mixture-of-Experts Autoencoders for Multi-Modal Deep Generative Models.

---

> > ### Comment · Reviewer_xSek · 2024-04-18
> > **Acknowledgement of response**
> >
> > Thank you for your response and clarifications.

---

### Review · Reviewer_ATvF · 2024-04-08

**Summary Of Contributions:**

This paper proposed two methods to improve the expressiveness when learning the latent variable model under the incomplete data. The two methods are: 1) a mixture model with finite number of mixtures to parameterize the variational posterior; 2) an imputation model that completes the observation first, and then parameterize the variational posterior given the completed data. The paper has performed experiments on synthetic data and real-world UCI dataset and minist and omniglot dataset. Compared to some existing baselines, the paper has shown better results.

**Audience:**

Yes

**Broader Impact Concerns:**

no concerns.

**Claims And Evidence:**

No

**Requested Changes:**

Please see the suggestions in the weakness section.

**Strengths And Weaknesses:**

Strength:

- learning latent variable model under the situation where the observed data is incomplete is an interesting task
- the two proposed methods are reasonable, though some of the details need to be further justified


Weakness:

- The relationship between the hierarchical latent variable model needs to be further justified.

The missing piece of the data itself can be the latent variable. In this aspect, we can either treat $\tilde{z} = [z, x_{mis}]$ and use the vanilla latent variable model framework like VAE for training. Or we can treat it in a hierarchical way, i.e., modeling both $p(x_{mis}|x_{obs})$ and $p(z|x_{obs}, x_{mis})$ and the factorization here incorporates the domain knowledge on the different roles of the two missing pieces ($x_{mis}$ and $z$).

- Regarding the mixture model, the optimization technique needs further justification.

In the mixture model formulation, if the number of mixtures is not large, why can't we simply do explicit integration (i.e., summation over K)? I'm not sure if the sampling is ever needed here. In this way we can also avoid back-propagating through non-differentiable categorical distribution.

- The experiments are mostly on toy datasets. Some more interesting datasets should be presented (e.g., coco with missing image segmentations, or imagenet with missing class labels)

Or some more interesting tasks where the missing data itself is not artificial.

- Stronger baseline methods need to be included.

Conceptually, the limitation of the assumption on the variational posterior (i.e., factorized gaussian) limits the expressiveness of the approximation. If we can include some baselines that have better expressiveness in the parameterization of variational posterior, then that would be more convincing.

---

> ### Author Response · Authors · 2024-04-12
> **Response (Part 1)**
>
> Thank you for your thoughtful feedback and suggestions. We respond to each question one-by-one below. In the updated manuscript, the updated text is highlighted using red colour.
>
> > * The relationship between the hierarchical latent variable model needs to be further justified.
> >
> > The missing piece of the data itself can be the latent variable. In this aspect, we can either treat $\tilde z = [z, x_{\text{mis}}]$ and use the vanilla latent variable model framework like VAE for training.
>
> This option has two issues. Firstly, if we assume a factorised distribution over $\tilde z = [z, x_{\text{mis}}]$, i.e., $q(\tilde z \mid x_{\text{obs}}) = \prod_j q(\tilde z_j \mid x_{\text{obs}})$, this is equivalent to assuming that the $\tilde z$ are conditionally independent given $x_{\text{obs}}$, which also means that the latents $z$ and the missing values $x_{\text{mis}}$ are conditionally independent. Conditional independence between the unobserved visibles $x_{\text{mis}}$ and the latents $z$ means that the latents do not carry any information about the missing visibles, which is generally undesirable as it leads to posterior collapse in VAEs [1]. Secondly, if the $x_{\text{mis}}$ are discrete (such as binarised MNIST and Omniglot data) you will need to use score-function gradient estimators, which can be high variance, hence needing additional work to reduce this variance to workable levels. The first issue is arguably the most critical here as it induces rather strong assumptions about the model.

---

> > ### Author Response · Authors · 2024-04-12
> > **Response (Part 2)**
> >
> > > Or we can treat it in a hierarchical way, i.e., modeling both $p(x_{\text{mis}} \mid x_{\text{obs}})$ and $p(z \mid x_{\text{obs}}, x_{\text{mis}})$ and the factorization here incorporates the domain knowledge on the different roles of the two missing pieces ($x_{\text{mis}}$ and $z$).
> >
> > Joint modelling of $p(z, x_{\text{mis}} \mid x_{\text{obs}})$ using variational inference with a distribution $q(x_{\text{mis}} \mid x_{\text{obs}})q(z \mid x_{\text{obs}}, x_{\text{mis}})$ is difficult. The difficulty here lies in specifying a flexible approximate distribution $q(x_{\text{mis}} \mid x_{\text{obs}}) \approx p(x_{\text{mis}} \mid x_{\text{obs}})$. This is mainly because of the following reason: the partition of the complete variable $x$ into observed $x_{\text{obs}}$ and missing $x_{\text{mis}}$ components in the training data can be arbitrary and different for each data-point. This means that there can be up to $2^D$ such partitions, one for each pattern of missingness. The challenge then boils down to specifying the distribution $q(x_{\text{mis}} \mid x_{\text{obs}})$ such that it can work with arbitrary variables in the conditioning and target sets. But it is not clear how to do this effectively (i.e. without inducing additional assumptions), which we highlight in the penultimate paragraph of Section 4.2. For example, if you were to use the "standard" conditional independence assumption on $q(x_{\text{mis}} \mid x_{\text{obs}})$, this will induce a bias to the model as the missing variables are generally not independent given the observed and hence should be avoided.
> >
> > Instead of approximating $p(x_{\text{mis}} \mid x_{\text{obs}})$ using variational inference, in Section 4.2 (DeMissVAE) we approximate this distribution using (approximate) samples, and denote it as $f^t (x_{\text{mis}} \mid x_{\text{obs}})$. In this way we avoid specifying a complicated distribution $q(x_{\text{mis}} \mid x_{\text{obs}})$, whereas the $q(z \mid x_{\text{obs}}, x_{\text{mis}})$ that DeMissVAE needs is learnt using standard VI with completed (or imputed) data using equation 12. Moreover, in the decoder objective (equation 10) we use the common assumption that we can marginalise the missing variables from the decoder distribution (because of the conditional independence of $x$ given $z$). Due to the marginalisation, the objective in equation 10 corresponds to an ELBO using the imputation-mixture $\mathbb{E}\_{ f^t(x_{\text{mis}} \mid x_{\text{obs}}) } [q(z \mid x_{\text{mis}}, x_{\text{obs}})]$ to approximate the incomplete data posterior $p(z \mid x_{\text{obs}}) = \mathbb{E}\_{ p(x_{\text{mis}} \mid x_{\text{obs}}) } [p(z \mid x_{\text{mis}}, x_{\text{obs}})]$. Since the imputations $x_{\text{mis}}$ only appear in the expectation, defining the distribution of the latents $z$, the objective mitigates potential degradations of the parameter estimation due to approximation errors in the imputation distribution $f^t(x_{\text{mis}} \mid x_{\text{obs}})$ (Appendix C).
> >
> > Alternatively, in Section 4.1, we choose a simpler approach, where the distribution of missing variables $p(x_{\text{mis}} \mid x_{\text{obs}})$ is never used. Instead, we approximate $p(z \mid x_{\text{obs}}) = \mathbb{E}\_{ p(x_{\text{mis}} \mid x_{\text{obs}}) } [p(z \mid x_{\text{mis}}, x_{\text{obs}})]$ directly with a finite mixture $\sum_{k=1}^K q(k \mid x_{\text{obs}}) q^k(z \mid x_{\text{obs}}))$. Here the components of the mixture may learn to represent the role of different imputations $x_{\text{mis}}$, but it is not explicitly enforced (unlike in Section 4.2).
> >
> > We hope that our comprehensive reply above answers your question. To summarise, we do not aim to jointly model the missing variables and the latents, because it is difficult to do so. Instead we model the incomplete-data posterior $p(z \mid x_{\text{obs}})$ using two different mixture-based approaches in Sections 4.1 and 4.2. Importantly, both approaches rely on the assumption that we can marginalise the missing variables from the decoder distribution (equation 1). In the updated manuscript (after equation 1) we now more explicitly highlight that we make use of this assumption.

---

> > > ### Author Response · Authors · 2024-04-12
> > > **Response (Part 3)**
> > >
> > > > In the mixture model formulation, if the number of mixtures is not large, why can't we simply do explicit integration (i.e., summation over K)? I'm not sure if the sampling is ever needed here. In this way we can also avoid back-propagating through non-differentiable categorical distribution.
> > >
> > > This is actually why we suggested the stratified objectives (SELBO and SIWELBO) in equations 5 and 8. These objectives do not require sampling the categorical distribution (as explained in the second paragraph after equation 5), but need to sample each component of the mixture and evaluate the decoder on each of the samples. Because of this, and as outlined in the discussion, the stratified approaches are more compute- and memory-expensive as they require evaluating the decoder at least once for each component of the mixture (i.e. at least K times), whereas the reparametrised approaches (equations 4 and 7) may work even with just a single sample from the whole mixture, but require implicit reparametrisation.
> > >
> > > In the updated manuscript, we now explicitly mention this in the discussion (Section 7) under "computational and memory budget" considerations.
> > >
> > > > * The experiments are mostly on toy datasets. Some more interesting datasets should be presented (e.g., coco with missing image segmentations, or imagenet with missing class labels).
> > >
> > > The COCO segmentation and Imagenet missing classes tasks are better solved using semi-supervised approaches. This is because in these tasks the whole mask/class is either missing or is observed, i.e. there is only one missingness pattern. Hence, there would be no issue in specifying the distribution $q(\text{class/segmentation} \mid \text{image})$, in contrast to the distribution we are dealing with, which has the form $p(x_{\text{mis}} \mid x_{\text{obs}})$, where $x_{\text{mis}}$ and $x_{\text{obs}}$ can be any arbitrary subsets of $x$. Hence, these tasks with only one pattern of missingness, i.e., semi-supervised tasks, are a sub-class of missing-data problems with simpler characteristics, which means that more efficient solutions may be obtained compared to the general missing-data case that we consider (for example, similar to the ones you have suggested above where we would simply model $q(\text{class/segmentation} \mid \text{image})$).
> > >
> > > The more general missing-data settings, where any number of variables may be missing for each data-point, are common in tabular/numerical data sets, which is why we have included the four UCI data sets that are commonly used to benchmark normalising flow models due to their complex interdependencies [2]. Moreover, the evaluation section includes experiments with 7 different methods, on 4 UCI data sets with 3 missingness fractions each, plus the MNIST and Omniglot data sets, hence a total of $(4 \cdot 3 + 2) \cdot 7 = 98$ experiments that are all repeated 5 times to obtain uncertainty estimates, not including the synthetic evaluations or the evaluations in the appendices.
> > >
> > > In the updated manuscript, we have rephrased the first paragraph in background (Section 2) explaining our setting more clearly---that the missing dimensions can be different for all data-points. This highlights the difficulty of our setting compared to the semi-supervised setting.

---

> > > > ### Author Response · Authors · 2024-04-12
> > > > **Response (Part 4)**
> > > >
> > > > > * Stronger baseline methods need to be included.
> > > > >
> > > > > Conceptually, the limitation of the assumption on the variational posterior (i.e., factorized gaussian) limits the expressiveness of the approximation. If we can include some baselines that have better expressiveness in the parameterization of variational posterior, then that would be more convincing.
> > > >
> > > > In this paper our goal was to re-use a variational family from the fully-observed case, which enables us to re-use the inductive biases from the well-studied scenarios with fully-observed data (Section 1, last paragraph; Section 3, last paragraph; Section 4 first paragraph). Using a specific family of distributions, whether the data is fully-observed or not, enables practitioners to induce desired structures in representations learnt with VAEs (i.e. inductive biases), which is important when, for example, we want to learn disentangled representations (see [3] for more examples). Hence, our evaluations did not involve the use of explicitly more flexible variational distributions to handle the complexities that arise due to missingnees.
> > > >
> > > > Moreover, the importance-weighted bounds (IWELBO) can be viewed to be _implicitly_ increasing the variational posterior expressiveness [4], which is included in our baselines, e.g. MIWAE that is hence a strong baseline. And indeed, we saw a relatively smaller increase in accuracy by using the mixture approach with the IWELBO-based bounds, compared to the ELBO setting. Although an additional reason for this may partially be due to the mode-seeking nature of the IWELBO/ELBO bounds, which can lead to mode collapse of the mixture components [5]. In the updated manuscript, we have extended the discussion in Section 7 to include this possibility.
> > > >
> > > > Finally, in the updated manuscript we have included additional experiments in Appendix F.3, where we have investigated what happens when the latent dimensionality is increased or decreased. This corresponds to increasing the expressiveness of the model and the variational distributions, which may be sufficient to handle the complexities stemming from missingness. The evaluation shows that the relative performance of the methods did not change significantly. The MissIWAE remains the best method on MNIST, and the MIWAE baseline is still slightly better than MissIWAE on Omniglot.
> > > >
> > > > References:
> > > >
> > > > [1] Shengjia Zhao, Jiaming Song, and Stefano Ermon. InfoVAE: Information Maximizing Variational Autoencoders.
> > > >
> > > > [2] George Papamakarios, Theo Pavlakou, and Iain Murray. Masked Autoregressive Flow for Density Estimation.
> > > >
> > > > [3] Ning Miao, Emile Mathieu, N. Siddharth, Yee Whye Teh, and Tom Rainforth. On Incorporating Inductive Biases into VAEs.
> > > >
> > > > [4] Chris Cremer, Quaid Morris, and David Duvenaud. Reinterpreting Importance-Weighted Autoencoders.
> > > >
> > > > [5] Yuge Shi, N. Siddharth, Brooks Paige, and Philip Torr. Variational Mixture-of-Experts Autoencoders for Multi-Modal Deep Generative Models.

---

### Decision · Action_Editor_KYhD · 2024-06-13

**Recommendation:** Accept with minor revision

**Comment:**

The two methods introduced in the paper have not been considered before for VAEs with incomplete data, to the knowledge of the reviewers and myself. Therefore, we believe that the paper is a valuable addition to the literature, despite its limited experiments.

The initial submission lacked some details, but the paper's writing has been improved during the rebuttal. However, I feel there is still room for improvement. In particular, the update of the imputation distribution is very unclear to me. On page 20, the authors mention methods such as pseudo-Gibbs (Rezende et al., 2014), Metropolis-within-Gibbs (MWG, Mattei & Frellsen, 2018a), and latent-adaptive importance resampling (LAIR, Simkus & Gutmann, 2023), but I did not understand their relevance since these are sampling algorithms. Furthermore, the target distribution is not specified. Therefore, I require that the authors significantly clarify this part of the paper before final acceptance.

**Audience:**

The paper is relevant since designing VAE for incomplete datasets is a challenging problem.

**Claims And Evidence:**

This paper proposes two new Variational Autoencoder methods for dealing with incomplete data. More precisely, the first observation made in the paper is the increase in the posterior and variational complexity in that particular setting.

To address this problem, two VAEs along with explicit ELBOs for their training are presented. The first VAE model consists in considering a family of finite mixture of Gaussian distributions as variational family. The second one consists in relying on imputation distributions to obtain approximate missing samples  $x_m$ given the observations $x_o$  from an adaptive conditional density $f^t$. The samples $x_m$ are then used to learn a variational density of the form $q_\phi(\cdot | x_o  ) = \mathbb{E}[q_\phi(z | x_o,x_m)]$ where the expectation is over $x_m \sim f^t( | x_o)$

The paper illustrates the performance of their approaches on synthetic data, real-world UCI datasets and minist/omniglot datasets. Their approaches in these experiments compare favorably with existing baselines.

---

> ### Author Response · Authors · 2024-06-27
> **Final author response**
>
> Thank you and the reviewers for thoughtful comments that helped improve our paper. In the camera-ready version we have further improved the overall clarity of the paper and addressed the AE's comments on page 20.